**Dual state/rainfall correction via soil moisture assimilation for improved streamflow**
**simulation: Evaluation of a large-scale implementation with SMAP satellite data**
**Yixin Mao[1], Wade T. Crow[2] and Bart Nijssen[1]**
1: Department of Civil and Environmental Engineering, University of Washington, Seattle, WA
2: Hydrology and Remote Sensing Laboratory, Agricultural Research Service, USDA, Beltsville,
MD
Corresponding author: Bart Nijssen (nijssen@uw.edu)

**Abstract**

Soil moisture (SM) measurements contain information about both pre-storm hydrologic states and within-storm rainfall estimates, both of which are required inputs for event-based streamflow simulations. In this study, an existing dual state/rainfall correction system is extended and implemented in the 605,000 km$^2$ Arkansas-Red River basin with a semi-distributed land surface model. The Soil Moisture Active Passive (SMAP) satellite surface SM retrievals are assimilated to simultaneously correct antecedent SM states in the model and rainfall estimates from the Global Precipitation Measurement (GPM) mission. While the GPM rainfall is corrected slightly to moderately, especially for larger events, the correction is smaller than that reported in past studies due primarily to the improved baseline quality of the new GPM satellite product. In addition, rainfall correction is poorer in regions with dense biomass due to lower SMAP quality. Nevertheless, SMAP-based dual state/rainfall correction is shown to generally improve streamflow estimates, as shown by comparisons with streamflow observations across eight Arkansas-Red River sub-basins. However, more substantial streamflow correction is limited by significant systematic errors present in model-based streamflow estimates that are uncorrectable via standard data assimilation techniques aimed solely at zero-mean random errors. These findings suggest that more substantial streamflow correction will likely require better quality SM observations as well as future research efforts aimed at reducing systematic errors in hydrologic systems.

## 1. Introduction

Accurate streamflow simulation is important for water resources management applications such as flood control and drought monitoring. Reliable streamflow simulation requires accurate estimates of pre-storm soil moisture (SM) that control the partitioning of infiltration and surface runoff during rainfall events, as well as longer-memory subsurface flow (Freeze and Harlan, 1969; Western et al., 2002; Aubert et al., 2003). Good streamflow simulations also require realistic rainfall time series estimates.

SM measurements contain information about both antecedent hydrologic states and within-storm rainfall events. With advances in the quality and availability of in-situ and satellite-measured SM products, researchers have started to explore the potential of using SM measurements to improve the estimates of both pre-storm SM and within-storm rainfall. For example, multiple studies have attempted to assimilate SM measurements to improve the representation of antecedent SM states in hydrologic models via Kalman-filter-based techniques (e.g., Francois et al., 2003; Brocca et al., 2010, 2012; Wanders et al., 2014; Alvarez-Garreton et al., 2014; Lievens et al., 2015, 2016; Massari et al., 2015; Mao et al., 2019). Other studies have explored the use of SM measurements to back-calculate within-storm rainfall or to correct existing rainfall time series products (e.g., Crow et al., 2011; Chen et al., 2012; Brocca et al., 2013; Brocca et al., 2014; Brocca et al., 2016; Koster et al., 2016).

In the past decade, so-called dual state/rainfall correction systems have been implemented that combine *both* SM state-update and rainfall correction schemes to optimally improve streamflow simulations (e.g., Crow and Ryu, 2009; Chen et al., 2014; Alvarez-Garreton et al., 2016). Specifically, SM measurements (typically from satellite observation) are used to simultaneously update model states and correct the (typically satellite-observed) rainfall time series product used to force the model. The updated antecedent states and corrected rainfall are then combined as inputs into a hydrologic model to produce an improved streamflow simulation (see Fig. 1 for illustration of the dual correction system). Past studies have suggested that such systems generally outperform either state-update-only or rainfall-correction-only schemes (Crow and Ryu, 2009; Chen et al., 2014; Alvarez-Garreton et al., 2016), with the rainfall correction contributing more during high-flow events and the state updating contributing more during low flow periods (also see Massari et al., 2018).

While these past studies were encouraging, they applied the dual correction system only
to catchment-scale, lumped hydrologic models. In this study, a semi-distributed land surface
model, the Variable Infiltration Capacity (VIC) model, is implemented instead. The VIC model,
compared to the previous lumped models, includes a more detailed representation of both energy
and water balance processes (Liang et al., 1994; Hamman et al., 2018). The macroscale grid-
based VIC also better matches the true spatial resolution of satellite SM measurements and
provides a means for correcting large-scale streamflow analysis. In addition, earlier dual
correction studies used previous-generation satellite products such as the Advanced
Scatterometer (ASCAT) satellite SM data, the Soil Moisture Ocean Salinity (SMOS) satellite
SM data and the Tropical Rainfall Measuring Mission (TRMM) precipitation data. Here, we use
newer data products from the more recent Global Precipitation Measurement (GPM) mission
(Hou et al., 2014) and the NASA Soil Moisture Active Passive (SMAP) mission (Entekhabi et
al., 2010). Both the SMAP and GPM products provide near-real-time measurements over much
of the global land surface, making them especially useful for regions with scarce ground-based
rainfall and SM observations.
The main objective of this study is to assess the effectiveness of such a dual correction
system to improve streamflow simulations using recent satellite SM and precipitation products.
To address this main objective, we introduce methodological advances. Specifically, we 1)
extended the system to provide a probabilistic streamflow estimate via ensemble simulation and
analysis techniques (note that past studies focused solely on deterministic improvement), 2)
updated the rainfall correction scheme to take full advantage of the higher accuracy and temporal
resolution of newer satellite data products, and 3) investigated the potential cross-correlation of
errors in the dual system, thus validating the theoretical basis of our analysis system. These
methodological contributions will be presented throughout the paper.
The remainder of this paper is organized as follows. Section 2 describes the dual
correction system and our novel methodological contributions, as well as the study domain,
hydrologic model, and datasets used. Results are presented in Sect. 3. Section 4 discusses our
results and identifies lessons learned, and Sect. 5 summarizes our conclusions.

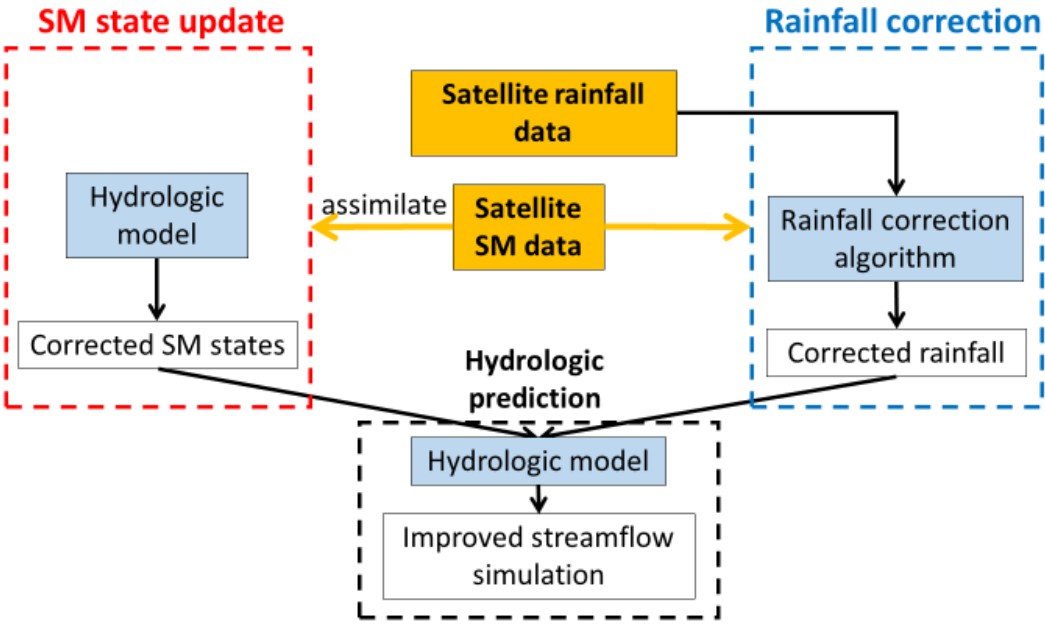


**Figure 1**. The dual state/rainfall correction framework applied in this study. Satellite-based soil

moisture (SM) data is integrated into a hydrological simulation system via two correction

schemes: 1) a standard data assimilation system to correct modeled SM states (shown in the red

box on the left), and 2) a rainfall correction algorithm to correct rainfall forcing data (shown in

the blue box on the right). Finally, these two contributions are combined to improve streamflow

simulations (shown in the black box at the bottom).

97

## 2. Methods

### 2.1. Study domain

The dual state/rainfall correction system is applied in the Arkansas-Red River basin

(approximately 605,000 km$^2$) located in the south-central United States (Fig. 2). This basin

consists of the Arkansas River and the Red River, both converging eastward into the Mississippi

River. This domain has a strong climatic gradient and is wetter in the east and drier in the west

(Fig. 2). The basin experiences little snow cover in winter except for the mountainous areas

along its far western edge. Vegetation cover tends to be denser in the east (deciduous forest) than

in the west (wooded grassland, shrubs, crops and grassland).

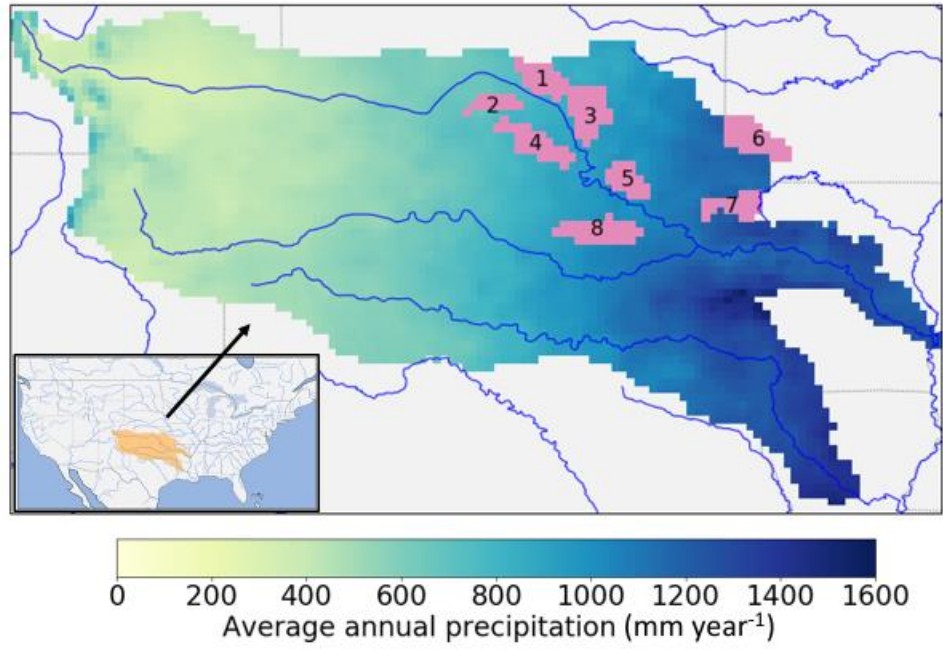

**Figure 2.** The Arkansas-Red River basin with climatology-averaged annual precipitation (calculated from NLDAS-2 precipitation data over 1979-2017). The pink shaded areas show the upstream sub-basins of the eight USGS streamflow sites evaluated in this study, with basin numbers labeled on the plot (see Table 1 for basin numbers and corresponding sites).

## 2.2. Data

### 2.2.1. SMAP satellite SM data

The SMAP mission provides SM estimates for the top 5 centimeters of the soil column, with an average revisit time of 2-3 days, a resolution of 36 km and a 50-hour data latency. Both ascending (PM) and descending (AM) retrievals from the SMAP L3 Passive product data Version 4 (O'Neill et al., 2016) from March 31, 2015 to December 31, 2017 were used in this study. A few SMAP pixels with obvious quality flaws (i.e., near-constant retrieval values) were manually masked out. The internal quality flags provided by the SMAP mission were not applied in this study to preserve the measurements in the eastern half of the domain, where the data quality of the entire region is flagged as unrecommended due to relatively heavy vegetation

cover. The native 36-km SMAP retrievals were used throughout the study without spatial
remapping or temporal aggregation.
**2.2.2 GPM satellite precipitation data**
The Integrated Multi-satellitE Retrievals for GPM (IMERG) Level 3 Version 05 Early
Run precipitation data was used in this study (Huffman et al., 2018). IMERG merges multiple
satellite observations and provides a near-global precipitation product with a spatial resolution of
0.1° (Huffman et al., 2015). We used the "Early Run" version of this product since its short
latency (4 hours) makes it suitable for near-real-time data assimilation applications. However,
this short latency also prevents correction of the IMERG "Early Run" product using ground-
based rain gauge observations. We aggregated the original 30-minute IMERG precipitation
product to our 3-hourly modeling time step and remapped it onto our 1/8° model spatial
resolution.
**2.2.3. Other meteorological forcing data**
Other than precipitation, the VIC model requires air temperature, shortwave and
longwave radiation, air pressure, vapor pressure and wind speed as forcing inputs. These
variables were taken from the 1/8° gridded North American Land Data Assimilation System
Phase 2 (NLDAS-2) meteorological forcing data product (Xia et al., 2009). We aggregated the
original hourly NLDAS-2 meteorological variables to the 3-hourly modeling time step.
**2.2.4. Validation data**
Daily streamflow data at eight USGS streamflow sites in the study domain (USGS, 2018)
was used to evaluate the streamflow time series from the dual correction system (Fig. 2 and
Table 1). These eight sites were selected for their lack of human regulation and their dense rain
gauge coverage (Crow et al., 2017). We separately evaluated the rainfall correction scheme, in
which the NLDAS-2 precipitation data was treated as the benchmark. The NLDAS-2
precipitation data was based on daily gauge-based rainfall measurements that were disaggregated
into hourly intervals using ground-based weather radar (Xia et al., 2012). NLDAS-2's reliance
on gauge observations (to obtain daily rainfall accumulations) ensures that it is more reliable (in
a relative sense) than the remote-sensing-only "Early Run" IMERG products used in this study.
Consequently, it provides an adequate evaluation benchmark for subsequent attempts to
correction IMERG.

**2.3. Hydrologic modeling**

We used Version 5 of the VIC model (Liang et al., 1994; Hamman et al., 2018). VIC is a

large-scale, semi-distributed model that simulates various land surface processes. In this study,
the VIC model was implemented in the Arkansas-Red River basin with the same setup as in Mao
et al. (2019). Specifically, the model was set up at 1/8º spatial resolution with each grid cell
further divided into multiple vegetation tiles via statistical distributions. Each grid cell was
simulated by VIC separately using a soil column discretized into 3 vertical layers (with domain-
average thicknesses of 0.10 m, 0.40 m and 0.93 m, respectively). In VIC, runoff can be generated
by fast-response surface runoff and by slow-response runoff from the bottom soil layer. All
vegetation cover and soil property parameters in the model were taken from Maurer et al. (2002),
which were calibrated against streamflow observations at the most downstream outlet of the
combined Arkansas and Red River basins. The simulation period was from March 2015 to
December 2017 when both the SMAP and GPM products are available. The VIC model was
spun-up by running the period 1979-2015 twice using NLDAS-2 forcing.

The local runoff simulated by VIC at each grid cell was routed through the stream

network using the RVIC routing model (Hamman et al., 2017), which is an adapted version of
the routing model developed by Lohmann et al. (1996, 1998).

**2.4. The dual correction system**

In this section, we describe our methodological updates to the rainfall correction scheme,

followed by a description of the state update scheme. Next, we describe how the two schemes are
combined to produce the final ensemble streamflow analysis.
**2.4.1. The SMART rainfall correction scheme updates and adaption**
The Soil Moisture Analysis Rainfall Tool (SMART) rainfall correction algorithm (Crow
et al., 2009; 2011; Chen et al., 2012) is based on sequential assimilation of SM measurements
into an Antecedent Precipitation Index (API) model:
$$API_t = \gamma API_{t-1} + P_t \tag{1}$$
where $t$ is a time step index; $P$ is the original IMERG precipitation observation [mm]; and $\gamma$ is a
unitless loss coefficient. We implemented a 3-hourly version of SMART (instead of the daily
version in past studies) to receive the 3-hourly IMERG rainfall input and both the ascending
(PM) and descending (AM) SMAP retrievals at the correct time of day. We also extended the
ensemble Kalman filter (EnKF) version of SMART introduced by Crow et al. (2011) to an
ensemble Kalman smoother (EnKS), in which the API state is not only updated at time steps
when SMAP is available, but also updated during measurement gaps (see Supplemental Material
Sect. S1 for mathematical details underlying the SMART EnKS approach). We set $\gamma$ to 0.98 such
that the un-corrected API time series approximately captures the dynamics of SMAP retrievals
(i.e., with high correlation; see Sect. S3 in Supplemental Material for a sensitivity analysis on $\gamma$).
SMAP was rescaled to the API regime through cumulative distribution function (CDF) matching
over the 2.5-year simulation period prior to assimilation. CDF matching was performed
separately for SMAP AM and PM retrievals to account for their mutual systematic differences.
The SMART algorithm then uses the API increment, $\delta_t$, to estimate the rainfall correction
amount via a simple linear relation. We implemented an ensemble rainfall correction rather than
the single deterministic rainfall correction used in past SMART applications:
$$P_{corr,t}^{(j)} = P_{pert,t}^{(j)} + \lambda \delta_t^{(j)} \tag{2}$$
where the superscript $(j)$ denotes the $j$th ensemble member (ensemble size $M = 32$); $P_{corr,t}$ is the
corrected precipitation for time $t$; $P_{pert,t}$ is the perturbed IMERG precipitation; and $\lambda$ is a scaling
factor that linearly relates API increment to rainfall correction, which was set to a domain-
constant of 0.1 [-] (see Supplemental Material Sect. S4 for discussion on the choice of $\lambda$). We
applied the rainfall correction only at timesteps when the original IMERG rainfall observation
was non-zero, taking advantage of the enhanced rain/no rain detection accuracy of IMERG
(Gebregiorgis et al., 2018). This tactic mitigates the spurious introduction of low intensity
rainfall events by SMART (see also Sect. 3.1). Finally, following Crow et al. (2009; 2011),
negative $P_{corr,t}$ values were set to zero, and the final corrected precipitation time series was
multiplicatively rescaled to be unbiased over the entire simulation period against the original
IMERG estimates (so that the long-term mean of the IMERG rainfall time series was preserved).

In this study, the SMART algorithm was run at each of the 36-km SMAP pixels

individually. The original 0.1° IMERG product was remapped to the coarser 36-km resolution
prior to SMART, and the corrected 36-km rainfall was then downscaled to the VIC 1/8° model
resolution. In our implementation of an EnKS-based SMART system, the original IMERG
precipitation was multiplicatively perturbed by log-normally distributed noise with mean and
standard deviation equal to one. SMAP measurement error ranges from 0.03 to 0.045 $m^3/m^3$
across the domain, which was estimated from the SMAP ground validation studies (e.g.,
Colliander et al., 2017; Chan et al., 2017), and its spatial distribution was set to be proportional
to leaf area index (LAI) (denser vegetation cover corresponds to larger SMAP error). The API
state was directly perturbed by zero-mean Gaussian noise to represent API model error. The
perturbation variance was set to 0.3 $mm^2$ over the entire domain such that the normalized filter
innovation has variance of approximately one (which is a necessary condition for the proper
parameterization of a Kalman filter; see Mehra (1971) and Crow and Bolten (2007)). The SMAP
measurement error and the state perturbation variance are the two primary variables impacting
innovation statistics. Since we had a relatively good estimate of the measurement error, the state
perturbation level can be uniquely determined via an analysis of normalized innovation variances
(Crow and van den Berg, 2010).

**2.4.2. State updating via EnKF**

As illustrated in Fig. 1 (the red box on the left), the SMAP SM retrievals were also

assimilated into the VIC model to update model states using an EnKF. The EnKF
implementation in this study generally follows Mao et al. (2019). Specifically, a 1D filter was
implemented for each 36-km SMAP pixel separately and at each pixel SMAP was assimilated to
update the SM states of multiple underlying finer 1/8° VIC grid cells. Resolution differences
between the coarser assimilation observations and finer modeling grid were accounted for via the
inclusion of a spatial averaging step within the observation operator (Mao et al., 2019).
Following Lievens et al. (2015; 2016) and Mao et al. (2019), only the upper two layers of SM
states in VIC were updated by the EnKF, although the bottom layer SM does respond to the
update of the upper two layers through drainage (see Sect. S2 in Supplemental Material for
mathematical details of the EnKF implemented here). An ensemble of 32 Monte Carlo model run
ensembles was used for the EnKF.
The SMAP retrievals were rescaled (separately for AM and PM retrievals) to match the
2.5-year mean and standard deviation of the VIC-simulated surface-layer SM time series prior to
assimilation. The error statistics of IMERG precipitation and unscaled SMAP retrievals were
assumed to be the same as those applied in SMART (Sect. 2.4.1). Following Mao et al. (2019),
VIC SM states were directly perturbed during the EnKF forecast step by zero-mean, additive
Gaussian noise with a standard deviation of 0.5 mm over the entire study domain. This noise
represents uncertainty in VIC's ability to propagate states estimates forward in time (note that the
bottom layer SM was perturbed, even though not directly updated by EnKF, to create a realistic
ensemble spread for probabilistic estimates of baseflow and, thus, streamflow).
Although VIC modeling errors are likely spatially auto-correlated, we tested whether
accounting for spatial correlation improved filter performance. Since it did not significantly
improve the results, we did not account for spatial correlation in our EnKF implementation. This
finding is consistent with Gruber et al. (2015) who described the limited benefit of 2-D filtering,
versus a 1-D baseline, when assimilating distributed SM retrievals into a land surface model. We
will further discuss this point in Sect. 4.
**2.4.3. Combining the state update and the rainfall correction schemes**
The ensemble of updated model states and the corrected rainfall forcing were combined
to produce final streamflow estimates (black box in the bottom of Fig. 1). We first randomly
paired ensemble members of corrected rainfall and updated VIC states and selected 32 such pairs
to balance competing considerations of computational cost and statistical stability. For each pair,
the VIC model was re-run with the updated states inserted sequentially over time and forced by
the corrected rainfall. Other meteorological forcings were kept unchanged. The runoff output
from VIC for each pair was then routed to the gauge locations, resulting in an ensemble of basin-
outlet streamflow time series. To further separate the relative contribution of the state update and
the rainfall correction schemes to overall streamflow improvement, two additional streamflow
simulations were performed. The first was the "state-updated streamflow" case, where VIC was
re-run with the updated states and forced by the original IMERG precipitation. The resulting
streamflow reflects only the impact of state updating on streamflow simulations. The second was
the "rainfall-corrected streamflow" case, where VIC was forced by the SMART-corrected
rainfall ensemble but without inserting the updated states. The resulting streamflow reflects only
the effect of SMART rainfall correction.
The EnKF state update and SMART rainfall correction schemes were executed
independently to minimize the risk of cross-correlated error (Crow et al., 2009). In particular,
note that VIC state estimates created using SMART forcing – see the black "Hydrologic
prediction" box in Fig. 1 – were not fed back into the EnKF state update analysis. Nevertheless,
cross-correlated error in (EnKF) state and (SMART) rainfall estimates potentially may still be
present since the two schemes are informed by the same SM measurement time series. Such
cross-correlated error could, in turn, degrade the quality of probabilistic streamflow estimates. In
fact, due to this concern, Massari et al. (2018) intentionally avoided combining the state and
rainfall correction schemes. To further investigate this risk, we performed a set of synthetic
experiments where we compared probabilistic streamflow estimates obtained via the following
two scenarios: 1) a single set of synthetically generated SM measurements assimilated into the
state and rainfall correction schemes, mimicking the original dual correction system; 2) two
separate sets of SM measurements with mutually independent errors assimilated separately into
the two schemes, thereby explicitly avoiding error cross-correlation in the system. Results show
that the two scenarios achieve very similar streamflow correction performance and, therefore,
minimal risk of degraded streamflow estimates (see Sect. S5 in Supplemental Material).

**2.5. Evaluation strategies and metrics**
We evaluated the rainfall correction results in addition to the dual-corrected streamflow
results in terms of both deterministic and probabilistic metrics.
The 1/8° gauge-informed NLDAS-2 precipitation data was remapped to the 36-km
SMART resolution grid as the benchmark for evaluating rainfall. Deterministically, the
ensemble-mean SMART-corrected rainfall was compared to the original IMERG precipitation
(remapped to 36 km), and its improvement was evaluated in terms of: 1) time series correlation
coefficient ($r$); 2) percent error reduction (PER) in terms of the root-mean-squared error

296 (RMSE); 3) additional categorical skill metrics, including false alarm ratio (FAR), probability of

297 detection (POD) and threat score (TS) (Wilks, 2011; Crow et al., 2011; Chen et al., 2012; Brocca

298 et al., 2016). Probabilistically, the normalized ensemble skill (NENSK) was calculated, which

299 measures the ensemble-mean error normalized by ensemble spread:

300        $NENSK = \dfrac{ENSK}{ENSP}$         (3)

301 where the ensemble skill (ENSK) is the temporal mean of ensemble-mean squared error, and the

302 ensemble spread (ENSP) is the temporal mean of ensemble variance (De Lannoy et al., 2006;

303 Brocca et al., 2012; Alvarez-Garreton et al., 2014; Mao et al., 2019). If an ensemble of time

304 series correctly represents the uncertainty of an analysis, NENSK will equal one (Talagrand et

305 al., 1997; Wilks, 2011). NENSK > 1 indicates an under-dispersed ensemble while NENSK < 1

306 indicates an over-dispersed ensemble. For all metrics, precipitation datasets were aggregated to

307 multiple temporal accumulation periods (the native 3-hour period without aggregation; 1-day; 3-

308 day) for evaluation at different time scales.

309   The dual-corrected streamflow was evaluated at the outlet of the eight USGS sub-basins

310 shown in Fig. 2. Deterministically, the ensemble-median corrected streamflow was compared to

311 the baseline streamflow, or the so-called "open-loop" streamflow, which is simply the single

312 VIC simulation forced by IMERG precipitation without any correction, in terms of 1) PER; and

313 2) the Kling-Gupta efficiency (KGE) (Gupta et al. 2009). The latter combines the performance of

314 correlation, variance and bias. Ensemble-median instead of ensemble-mean streamflow was used

315 for more stable evaluation results in the case of a skewed streamflow ensemble caused by model

316 nonlinearity. In addition to ensemble-median evaluations, NENSK was calculated for the entire

317 streamflow ensembles.


319 **3. Results**

320 **3.1. SMART rainfall correction**

321 **3.1.1. The impact of SMART methodological choices**

Figure 3 shows the rainfall improvement in terms of correlation coefficient $r$ based on
both an EnKS- (the left column) and EnKF-based (the right column) implementation of SMART.
For EnKF results, both $\delta$ and $P$ in Eq. (2) were aggregated to 3-day windows prior to correction
to ensure SM data availability in every correction window (and the 3-day correction was
subsequently downscaled to 3-hour time steps uniformly). Overall, the EnKF implementation
results in less $r$ improvement than the EnKS implementation, which confirms the benefit of
applying SMART using a smoothing approach.
The impact of our (previous choice) to update rainfall only at non-zero IMERG time
steps is examined via domain-median categorical metrics (Fig. 4). When we correct rainfall
every time step (Fig. 4 Column 1), FAR is largely degraded (by $0.1 - 0.4$) at low rainfall event
thresholds especially with shorter accumulation periods (3-hour and 1-day; see Fig. 4a). This is
likely due to SMART misinterpreting SM retrieval noise as small rainfall events (Chen et al.,
2014). POD is improved at these low thresholds (Fig. 4b), but not enough to compensate for the
large FAR degradation. Therefore, TS, which accounts for both false alarms and missed events,
is also degraded at low thresholds (by as large as 0.2 at 3-hourly). In contrast, when we only
correct rainfall at non-zero IMERG time steps (Fig. 4 Column 2), the FAR degradation is much
less (note the different y-axes in the two columns in Fig. 4). While this approach does sacrifice
POD at low thresholds (Fig. 4e), the overall TS for 1-day and 3-day aggregation is improved for
most event thresholds, especially the higher ones. As mentioned in Sect. 2.4.1, one possible
reason for the success of this SMART choice is the improved rain/no rain detection quality of the
baseline IMERG precipitation product, which was found to have improved miss-rain, false-rain
and hit rate relative to older TRMM TMPA-RT products over the Continental U.S. (Gebregiorgis
et al., 2018). It is thus beneficial to retain IMERG's rain/no rain detection skill and not subject it
to SMART-based correction.
With regards to binary rain/no-rain determination, one strategy for mitigating FAR
problems is to arbitrarily set a (greater than zero) minimum accumulation threshold that must be
exceeded for an event to be registered. To this end we carried out a sensitivity analysis to
examine the impact of using a non-zero rain/no rain threshold versus our baseline assumption of
a zero threshold. However, this analysis was unable to isolate an optimized threshold value for
distinguishing rain/no rain cases. Instead, a continuous trade-off exists between POD and FAR at

different rainfall thresholds. However, a zero rain/no rain threshold does appear slightly

beneficial for PER and the correlation coefficient improvement (see Sect. S6 in Supplemental

Material).

### 3.1.2. Rainfall correction evaluation

After rainfall correction at 1-day and 3-day accumulation periods, PER exhibits a

domain-median error reduction of ~8% (Fig. 5 Column 1). The PER improvement is consistent

with the improvement of the categorical metrics at high-event thresholds (Fig. 4 Column 2),

since PER is more sensitive to high rainfall values. Three-hourly PER shows little improvement

(Fig. 5a), suggesting that the deterministic correction is more effective at an accumulation period

that more closely matches the SMAP retrieval interval. The same finding can also be drawn from

the correlation and categorical results (Fig. 3 Column 2 and Fig. 4 Column 2). Overall, the

correlation coefficient improves more in the western part of the domain, which is likely

attributable to the better quality of SMAP retrievals in the lightly vegetated western portion of

the basin. However, RMSE is reduced more in the eastern part of the domain, which is likely due

to the increased frequency of large rainfall events in this region, and SMART's tendency to be

more effective for the correction of moderate-to-large precipitation events. Note that SMART

rainfall correction cannot be evaluated in terms of overall bias, since – like all SM data

assimilation systems - the SMART algorithm rescales the corrected time series back to the

uncorrected mean prior to its evaluation.

The probabilistic metric NENSK (Fig. 5 Column 2) is less than one for most of the

domain at a 3-hour time step, indicating an over-dispersed ensemble on average. However, when

evaluating at 1-day and 3-day accumulation periods, NENSK is closer to one, indicating a better

representation of the uncertainty of the rainfall estimates. As we aggregate over longer

accumulation windows (e.g., 3-day), NENSK becomes slightly greater than one (i.e., under-

dispersed ensemble), since the SMART algorithm assumes only a random rainfall error but no

systematic bias. As a result, it slightly underestimates the uncertainty range over longer-term

periods. Ensemble rainfall tends to be under-dispersed on the west edge of the domain with low

rainfall, indicating that we are underestimating rainfall uncertainty in this region.

In summary, SMART successfully uses SMAP SM retrievals to correct IMERG rainfall

during relatively larger events, with slight to moderate deterministic improvement. However,

SMART correction is less successful for small rainfall events and can even lead to slight
degradation. The correction is more effective, and the ensemble representation is better, when
rainfall estimates are temporally aggregated to periods consistent with SMAP retrieval intervals
(i.e., 1-day to 3-day accumulation periods).

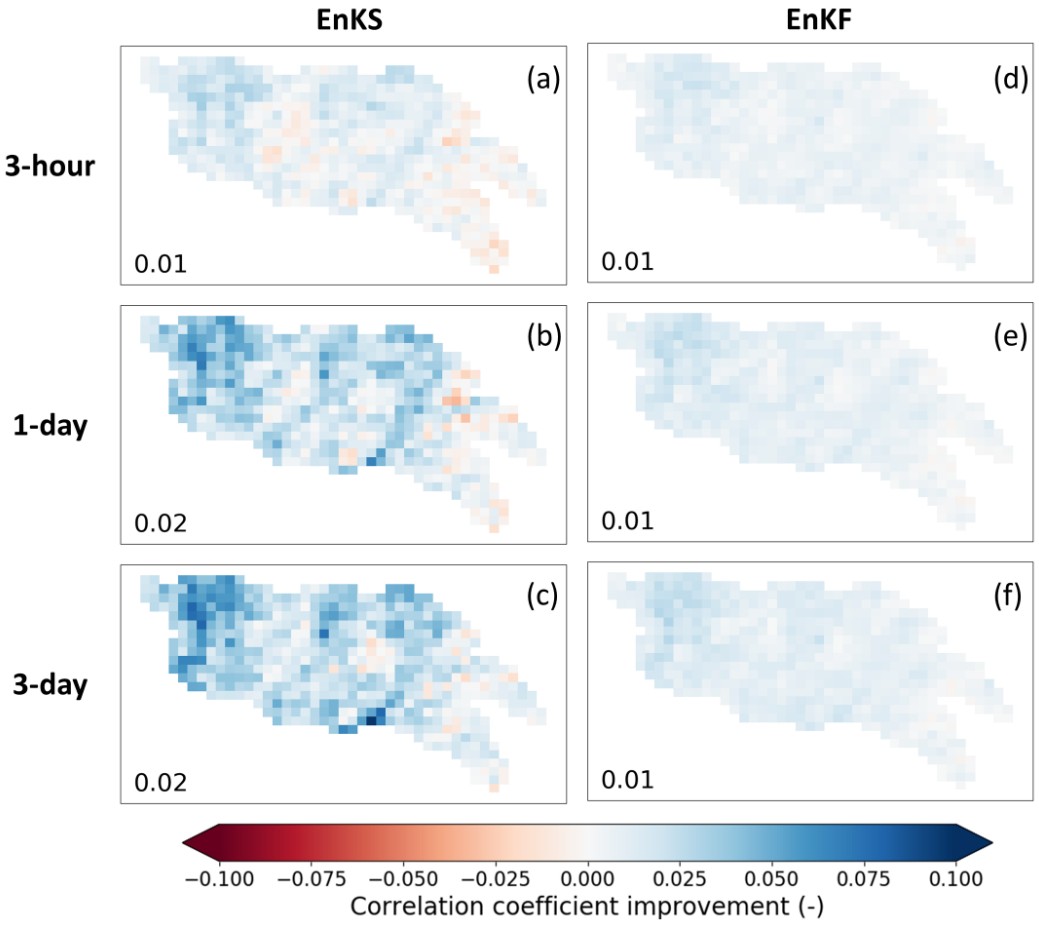


**Figure 3.** Maps of correlation coefficient improvement after SMART rainfall correction (i.e.,
improvement of correlation with respect to NLDAS-2 benchmark rainfall realized upon
implementation of SMART). The left column shows the SMART EnKS experiments (*a*, *b*, *c*),
and the right column shows the EnKF experiments (*d*, *e*, *f*). Each row shows results based on
different temporal accumulation periods (i.e., 3-hourly, 1-day and 3-day aggregation,
respectively). The number on the lower left corner of each subplot shows the domain-median
correlation improvement.

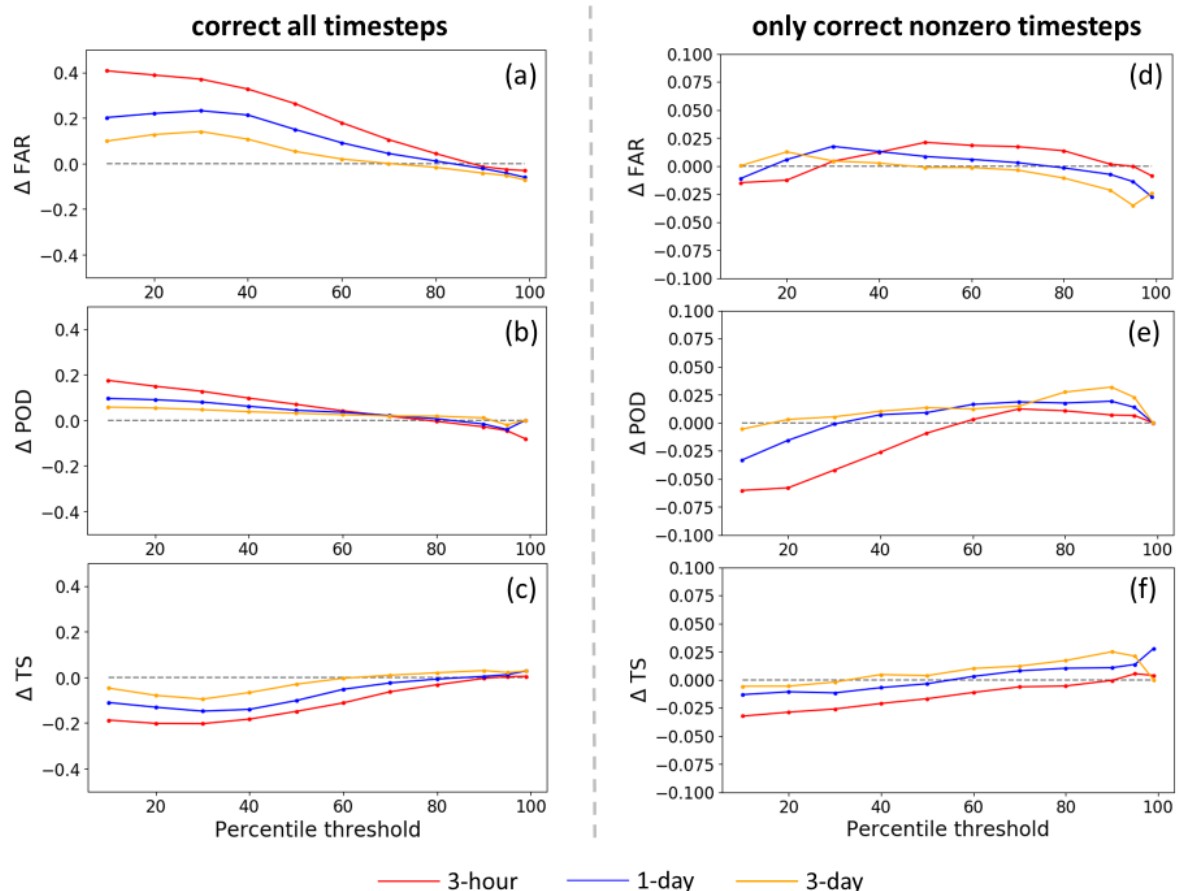


**Figure 4.** Change in categorical metrics (FAR, POD and TS) before and after SMART correction for 3-hourly, 1-day and 3-day accumulation periods. Metrics at different rainfall thresholds are shown on the *x* axis (e.g., the 80th percentile means that an event is defined as exceeding the 80th percentile of non-zero rainfall accumulation over the listed time accumulation period). The left column (*a*, *b*, *c*) is for SMART with rainfall corrected at all time steps; the right column (*d*, *e*, *f*) is for SMART with rainfall corrected only at non-zero time steps. Note that the y-axis range is different for the two columns.

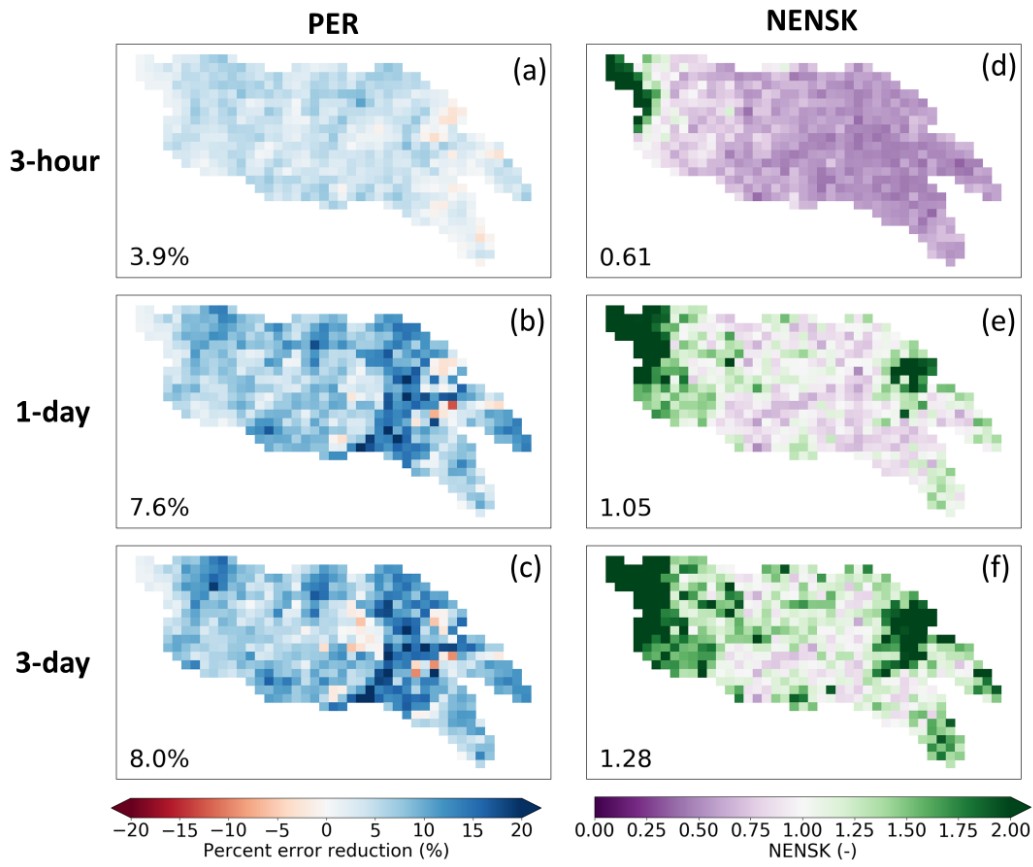

**Figure 5.** Maps of SMART rainfall correction results (with $\lambda = 0.1$, EnKS, and rainfall corrected only during time steps with non-zero rainfall). Each column shows the following metrics, respectively: percent RMSE reduction (PER) (*a*, *b*, *c*), and ensemble NENSK (*d*, *e*, *f*). Each row shows results based on different temporal accumulation period: 3-hourly, 1-day and 3-days, respectively. The number in the lower left corner of each subplot shows the domain-median statistic.

## 3.2. Streamflow from the dual correction system

### 3.2.1. Evaluation of streamflow improvement

The final daily streamflow performance from the dual correction system is listed in Table 2 (the "*dual*" columns) for each sub-basin. Overall, streamflow estimates are improved but with large variability across sub-basins. Specifically, PER ranges from approximately 6% to 34% and KGE improvement ranges from slightly negative to +0.95 across all sub-basins. For sub-basins

with better baseline streamflow performance (as measured by KGE, i.e., the Ninnescah, Walnut
and Chikaskia sub-basins), the relative improvement after the dual correction is generally
smaller.
Table 2 also summarizes the streamflow improvement from each of the correction
schemes alone (i.e., the "*state update only*" and "*rainfall correction only*" columns). For sub-
basins with relatively better open-loop model performance, the contribution of state updating
generally surpasses that of rainfall correction. Conversely, at sub-basins with relatively poorer
open-loop model performance (i.e., the Bird, Spring, Illinois and Deep sub-basins), streamflow
improvement is primarily attributable to the SMART rainfall correction.
**3.2.2. Impact of rainfall forcing error**
To further understand the relationship between open-loop simulation performance,
rainfall forcing error and correction performance, we forced the VIC model by the NLDAS-2
benchmark rainfall (without state update). The subsequent streamflow improvement level is
assumed to approximate the maximum improvement achievable via rainfall correction alone
(Table 2 "*NLDAS2-forced*" columns). While almost all sub-basins show streamflow
improvement simply by switching to NLDAS-2 rainfall forcing, the improvement is especially
large for sub-basins with poorer open-loop streamflow estimates. In these basins, PER is over
65% and the negative KGE for the open loop case improves to near zero or positive values for
the NLDAS-forced case. This suggests that, despite advances in the quality of remotely sensed
rainfall data products, poor open-loop streamflow simulations at these sub-basins are still largely
attributable to poor-quality IMERG rainfall forcing error. In these basins, SM-based rainfall
correction scheme can potentially play an important role in improving VIC streamflow estimates.
Unfortunately, this potential is not always realized. Note how the SMART-based rainfall-
correction-only case generally fails to match NLDAS-forced case in the Spring, Illinois and
Deep sub-basins (Table 2). This is likely because these basins are located in relatively high
biomass areas where SMAP retrievals (and thus SMART corrections) are less accurate.
In contrast, the sub-basins with better open-loop streamflow results (i.e., the Ninnescah,
Walnut and Chikaskia sub-basins) demonstrate less streamflow improvement when switching to
the NLDAS-2 rainfall forcing. The sub-basin with best (IMERGE-forced) open-loop streamflow
results, Chikaskia, even experiences a small degradation when forced by the NLDAS-2 rainfall
(Table 2). This suggests that the NLDAS-2 benchmark rainfall at this sub-basin is not obviously
superior than the IMERG baseline. Nevertheless, SMART is still able to extract information
from SMAP and slightly correct IMERG rainfall and subsequent streamflow estimates.

**3.2.3. Impact of model parameterization**

The dual correction scheme presented in this study is designed to correct only the random
error present in a hydrologic simulation system. It does not correct systematic error or overall
bias. Figure 6 shows example time series of the open-loop, USGS-observed and dual-corrected
streamflow at three sub-basins (the Chikaskia, Deep and Illinois) with various levels of open-
loop performance. Although the dual system often nudges the simulated streamflow in the
correct direction (especially during high-flow periods) and results in overall improved evaluation
statistics, systematic error (in the model process representation as well as rainfall forcing) clearly
exists. This systematic error, although difficult to quantify, cannot be corrected by the data
assimilation approach discussed here. The NENSK statistic partly reflects such systematic error.
NENSK is significantly above one at most sub-basins, indicating an under-dispersed ensemble
on average. In other words, at most sub-basins the ensemble spread created by the dual system
only represents the random uncertainty around the open-loop streamflow and neglects systematic
error that accounts for a significant fraction of total streamflow error.
The level of systematic error is tied closely to the quality of the hydrologic model
parameters often estimated through calibration. The VIC parameters used in this study were
taken from Maurer et al. (2002) and derived based on streamflow at the outlets of large basins.
To further examine the effect of systematic error on data assimilation, we calibrated the model
parameters for the eight sub-basins separately using streamflow acquired from the USGS (Table
1). Specifically, VIC parameters that control infiltration, soil conductivity and baseflow
generation as well as the recession rate of the grid-cell-scale unit hydrograph in RVIC were
calibrated using the MOCOM multi-objective autocalibration method (Yapo et al., 1998). Basin-
constant parameters were calibrated toward USGS streamflow time series during 2015 to 2017
(forced by the baseline IMERG precipitation) to optimize daily KGE and monthly bias. Only a
subset of the eight sub-basins achieved better-than-open-loop streamflow results via this
traditional calibration method, due mainly to the relatively large IMERG forcing error present in
some sub-basins that prevents the calibration scheme from finding an improved
parameterization. Figure 7 shows three example sub-basins (i.e., Chikaskia, Deep and Illinois)
with relatively good calibration outcomes. Comparing Fig. 7 to Fig. 6, we observe that the
streamflow improvement achieved by parameter calibration (i.e., systematic error reduction)
alone is as, or more, important than that achieved by data assimilation (via random error
reduction) in all three sub-basins. In both cases (i.e., the default and calibrated VIC parameters),
NENSK is significantly above one, indicating that we underestimate the streamflow simulation
uncertainty when only random errors are considered.

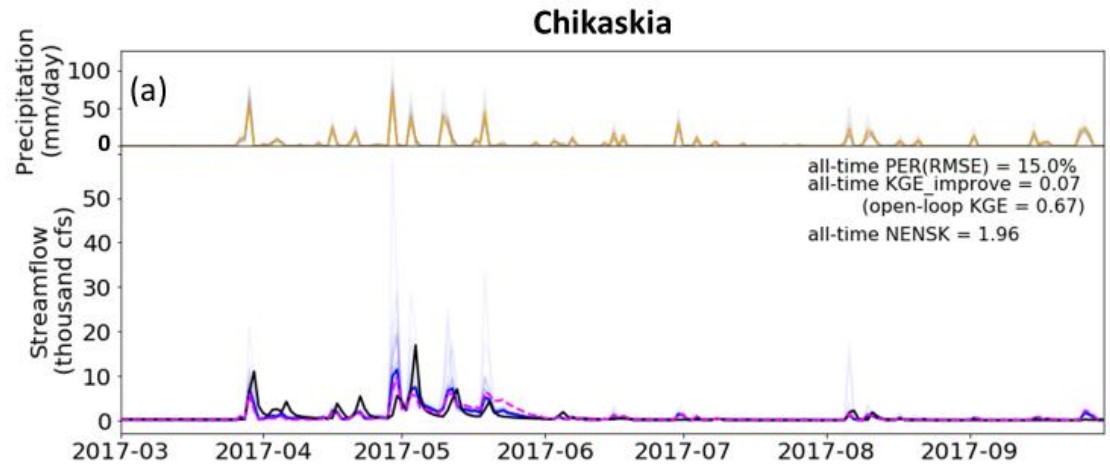

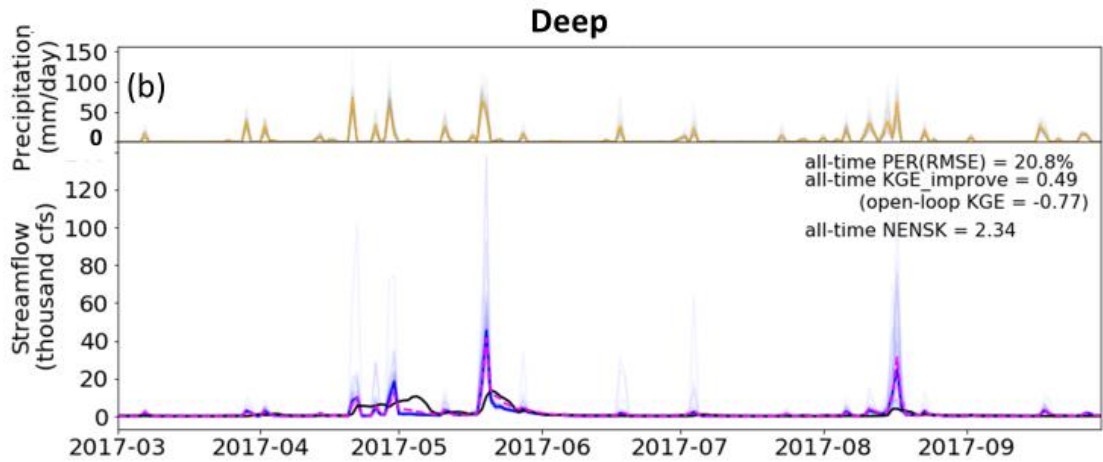

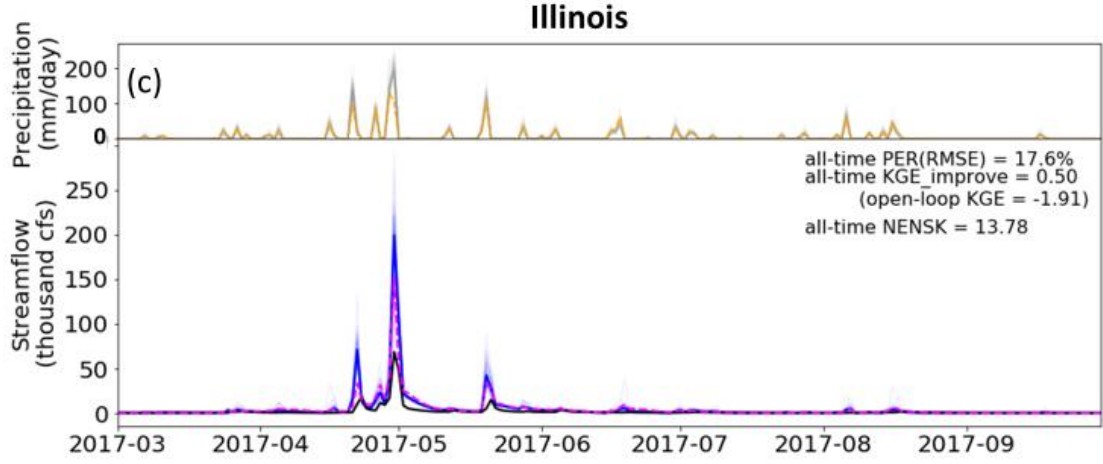

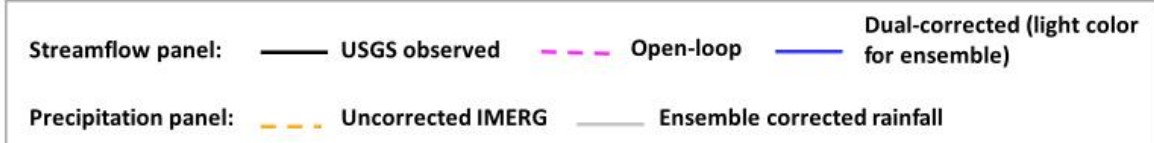

**Figure 6.** Example time series of streamflow results from the dual correction system. In the
lower panel, b*lack line*: USGS observed streamflow; *magenta line*: baseline VIC simulation;
*light blue lines*: ensemble updated streamflow results; *solid blue line*: ensemble-mean updated
streamflow. In the upper panel, *orange line*: uncorrected IMERG rainfall aggregated to the sub-
basin-average; *light grey lines*: ensemble corrected rainfall. Only part of the simulation period is
shown for clear display; however, statistics shown on each panel are based on the entire
simulation period (approximately 2.5 years).

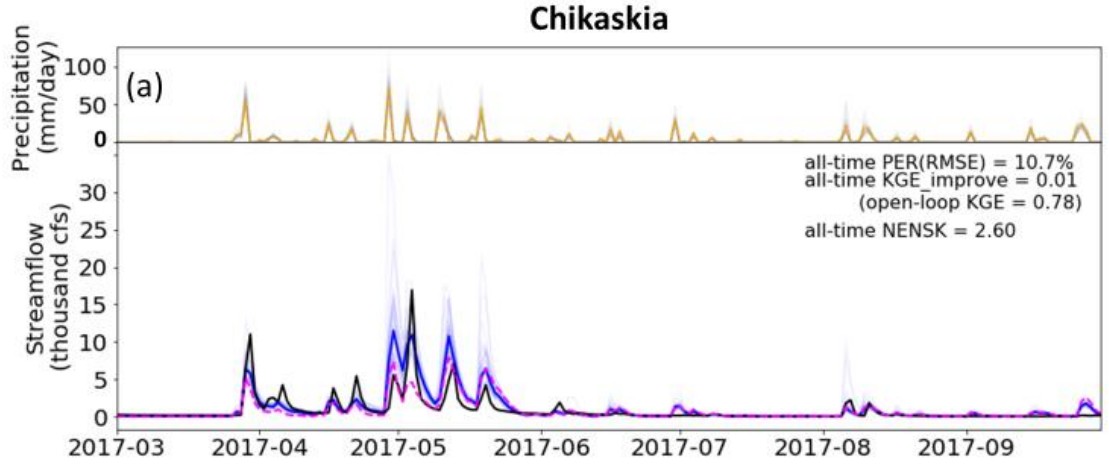

**Chikaskia**

(a)

all-time PER(RMSE) = 10.7%
all-time KGE_improve = 0.01
(open-loop KGE = 0.78)

all-time NENSK = 2.60

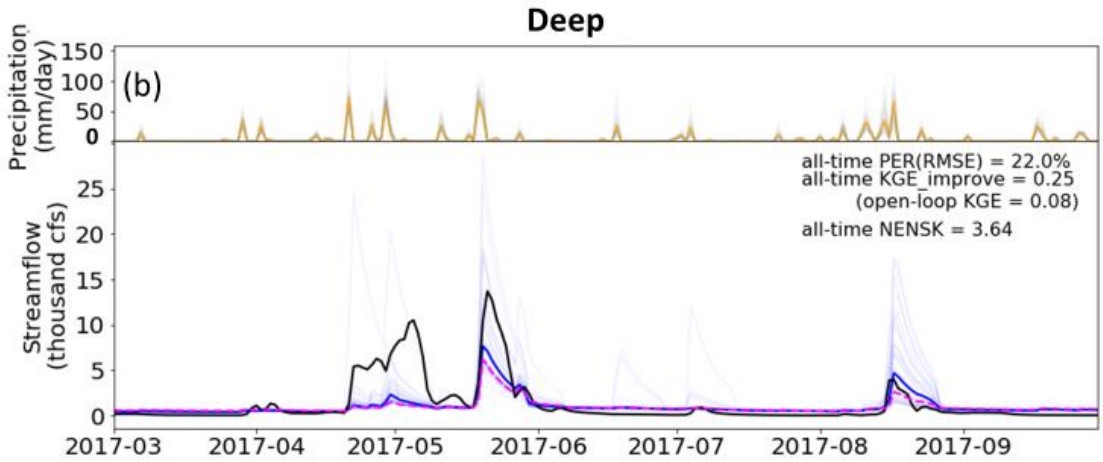

**Deep**

(b)

all-time PER(RMSE) = 22.0%
all-time KGE_improve = 0.25
(open-loop KGE = 0.08)

all-time NENSK = 3.64

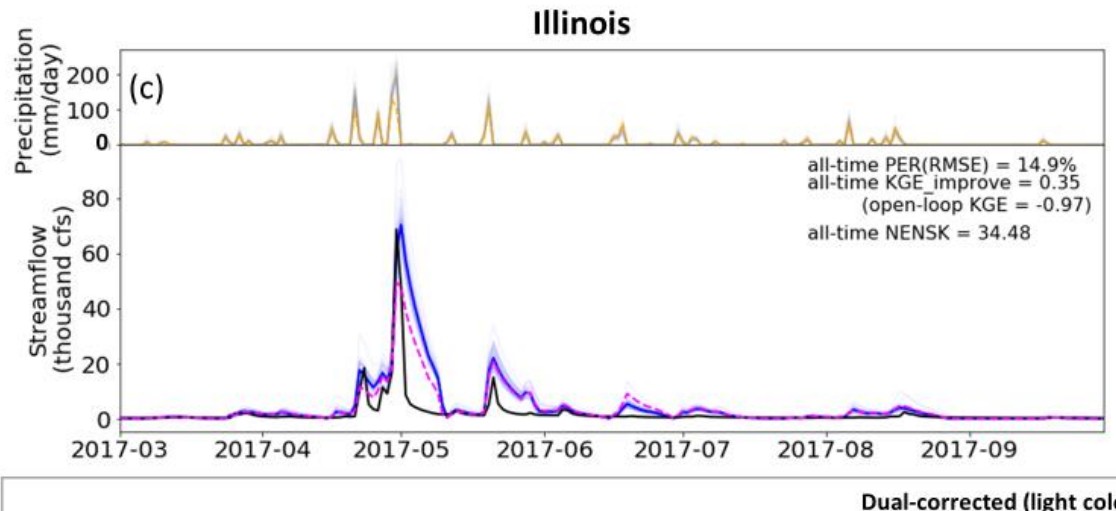

**Illinois**

(c)

all-time PER(RMSE) = 14.9%
all-time KGE_improve = 0.35
(open-loop KGE = -0.97)

all-time NENSK = 34.48

| Streamflow panel: | —— USGS observed | – – Open-loop | **Dual-corrected (light color for ensemble)** |

| Precipitation panel: | – – Uncorrected IMERG | —— Ensemble corrected rainfall |


**Figure 7.** Same as Fig. 6, but calibrated VIC model parameters.

## 4. Discussion

### 4.1. SMART rainfall correction

Overall, SMART improves the IMERG rainfall product (see Figures 3 to 5); however, the magnitude of improvement is somewhat smaller than that found in previous studies, especially in terms of correlation $r$ (domain-median improvement of 0.01 to 0.02). Table 3 summarizes results from past studies that applied remotely sensed SM to correct rainfall time series. Over the past decade, the quality of the baseline satellite-derived rainfall product has improved considerably, from the TRMM 3B40-RT product used by Crow et al. (2009) and Crow et al. (2011) with $r =$ ~0.5, to the TRMM 3B42-RT product used by Brocca et al. (2016) with $r =$ ~0.6 – 0.7, to the IMERG product used in our study with $r$ over 0.8. This tendency is confirmed by Gebregiorgis et al. (2018) who demonstrated the improved accuracy of IMERG relative to TRMM over the Continental U.S. in terms of correlation, RMSE, bias and categorical metrics. This improvement is relevant here because the marginal value of data assimilation tends to decrease as the skill of the background land surface model increases (Reichle et al., 2008; Qing et al., 2011; Bolten and Crow, 2012; Dong et al., 2019). Since SMART is fundamentally a data assimilation approach, the added value of its SM-based correction tends to *decrease* as the accuracy of the baseline product (it is correcting) increases. This tendency, previously noted in Crow and Ryu (2009) and Crow et al. (2011), is clearly illustrated in Table 3. Therefore, large improvement over time in the quality of satellite-based rainfall products appears to have partially undercut the value of SM-based rainfall correction. It should be noted that the SM/rainfall correction algorithms applied in Table 3 differ slightly. However, Brocca et al. (2016) found comparable performance even when inter-comparing very different rainfall correction approaches, suggesting that the various studies listed in Table 3 are relatively inter-comparable.

### 4.2. Dual correction for streamflow

Although we applied the dual correction system to the entire Arkansas-Red basin, we
selected only eight smaller sub-basins for our streamflow evaluation due to the limited
availability of unregulated streamflow observations at basin outlets. While the dual correction
approach generally improved VIC streamflow estimates, especially during relatively high flow
events in areas with poor IMERG data, the magnitude of this correction was relatively modest.
Results in Sect. 3 indicated three general reasons for this. First, the latest generation of satellite
rainfall products (e.g., IMERG) has significantly improved precision compared to its
predecessors. The already high-quality rainfall estimates are more difficult for SM retrievals to
contribute substantial rainfall correction skill (see discussion in Sect. 4.1 above). Second, the
dual correction approach is designed to correct only the zero-mean random error component in
the total streamflow error but not systematic error or bias. However, systematic error sources,
typically associated with inaccurate model structure and/or parameterization and large rainfall
bias, can account for a significant fraction of overall streamflow error (Sect. 3.2.3). The
existence of systematic error is particularly problematic from a probabilistic perspective, since
the ensemble streamflow produced by the dual system only represents random error, and
therefore largely underestimates simulation uncertainty. Finally, in certain sub-basins (i.e., the
Bird, Spring, Illinois and Deep sub-basins) where VIC streamflow is substantially degraded by
random error in IMERG data products, SMART-based dual correction often underperformed due
to the reduced accuracy of SMAP-based rainfall correction in eastern areas of the domain with
relatively dense biomass (see Fig. 3).
In addition to these factors, additional research is needed to fully investigate the impact
of several simplifications applied in the dual correction data assimilation system. For example,
the impact of error spatial correlation on downstream streamflow performance should be fully
examined before extending our findings to large-scale river systems. Specifically, while a 1-D
filter with spatially uncorrelated model representation error may be appropriate for small-basin
correction, ignoring the spatial correlation structure of errors could potentially have a more
profound impact on the correction performance at large river outlets where streamflow originates
from runoff from a large number of grid cells. Multiple studies have investigated the effects of
spatial error patterns in hydrologic data assimilation. For example, Reichle and Koster (2003)
investigated the impact of spatial error correlation in the model SM states on its assimilation
performance; Gruber et al. (2015) examined the impact of a 2-D filter with spatially auto-
correlated error versus a 1-D filter on SM updating quality; Pan et al. (2009) and Pan and Wood
(2009; 2010) evaluated the surface SM assimilation performance with VIC by comparing a 1-D
filter, a 2-D filter and a multiscale autoregressive filtering approach, as well as considering
spatial and temporal structure of precipitation error. However, all these studies focused
exclusively on the performance of SM simulations. Direct assessment of the impact of spatial
error patterns on the routed streamflow results is needed, especially from a probabilistic
perspective since the ignorance of spatial error patterns (and therefore their potential to mutually
cancel as runoff is routed through a river network) will lead to an incorrect ensemble
representation of streamflow uncertainty.
Another factor that may have limited the dual correction performance, particularly the
state updating scheme, is the rescaling of the SMAP retrievals to the VIC top-layer SM regime.
Matching a satellite-observed SM product with that represented in a land surface model (LSM) is
a necessary pre-processing step in a data assimilation system; however, it has the well-known
limitation of neglecting potential bias-correction information contained in the satellite-observed
product. While this problem is well-discussed in the literature (see, e.g., Yilmaz et al., 2013;
Kumar et al., 2015; Nearing et al., 2018), no robust solutions currently exist. Ideally, the physical
source of remote sensing and modelling biases could be isolated and addressed. However, this is
very difficult to do in practice. For instance, although SMAP is typically described as measuring
the top ~ 5 cm of SM, the actual vertical support depth is unclear and varies nonlinearly as a
function of SM and vegetation water content. In addition, the relationship between the top-layer
depth and its SM dynamics in an LSM is complex and driven by multiple poorly known model
parameters (although, Shellito et al. (2018) found that changing the top-layer depth from 10 cm
to 5 cm in the Noah LSM did not significantly affect surface SM dynamics). Therefore, like
other existing SM data assimilation applications, we are forced to resort to an ad hoc solution
where satellite-based observations are rescaled to match the climatological characteristics of
equivalent model products.

**5. Conclusion**
In this paper, we applied a dual state/rainfall correction data assimilation system in the
Arkansas-Red River basin. Built upon the dual system developed in past studies, we have made
several methodological advances. First, we implemented the dual correction system with a more
complex, semi-distributed land surface model (VIC) and applied it in a regional-scale basin.
Second, the latest satellite products, the SMAP SM product and the IMERG rainfall product,
were incorporated into the system. Third, the existing dual correction algorithm was extended to
maximize the use of information contained in the more accurate, and temporally more frequent,
satellite data products. Fourth, the SMART approach has been modified to produce an ensemble
streamflow product to generate probabilistic estimates. Fifth, we confirmed via a formal
synthetic experiment that error cross-correlation that potentially exists in the dual correction
system does not cause noticeable degradation of streamflow improvement and the dual
correction scheme applied here is optimal.
Our results show that, overall, the SMART algorithm is able to correct IMERG rainfall
slightly to moderately, and the correction is more effective during larger rainfall events and at
daily to multi-daily time scales. The ensemble produced by the correction scheme represents the
rainfall uncertainty relatively well. However, the rainfall correction we achieved is generally
smaller than found by previous studies, mainly due to improved quality of the baseline satellite
rainfall product over time. In addition, although SMAP arguably also has higher quality than
older remotely-sensed SM products, its quality remains relatively low in dense-biomass regions,
resulting in reduced rainfall correction via SMART.
Combined with analogous improvement in pre-storm SM states, the relatively small
rainfall correction is propagated into VIC and generally results in improved streamflow
estimates. However, the improvements found are relatively small and vary greatly between sub-
basins. Due to its deleterious impact on SMAP retrieval uncertainty, small improvement is found
in sub-basins containing dense biomass. Furthermore, the dual data assimilation system is only
designed to correct zero-mean random errors and not systematic errors or bias. However,
systematic errors can account for a substantial fraction of the total streamflow error. This results
in relatively modest streamflow correction via the Kalman-filter-based correction system and the
significant underestimation of uncertainty in VIC streamflow estimates.
Given the above findings, we provide the following recommendations for future
research:

1) Higher-quality SM retrievals are necessary to push the current limit of rainfall

correction (and, consequently, streamflow correction) especially in areas of dense vegetation.

2) However, even with better SM data quality, data assimilation techniques aimed solely

at random error sources are unlikely to substantially reduce streamflow errors in many sub-
basins, since random errors often account for only a relatively small portion of the total error.
Instead, approaches that reduce systematic errors in streamflow simulation are needed. To date,
this is still a challenging task in large-scale hydrologic modeling, since robust calibration is
difficult to achieve with limited streamflow data and many distributed parameters. With the
availability of the near-global and distributed satellite products such as SMAP and IMERG, more
creative methods are needed to extract useful information from the large volume of remote
sensing observations. For example, the characteristics of SM dynamics and its response to
rainfall can be directly extracted from the datasets themselves, which can potentially inform
hydrologic model representation. These new areas of research have the potential to improve
hydrologic modeling beyond the correction of random errors.

**Code availability**

The VIC model used in the study can be found at https://github.com/UW-Hydro/VIC.

Specifically, we used VIC version 5.0.1 (doi:10.5281/zenodo.267178) with a modification to the
calculation of drainage between soil layers (https://github.com/UW-
Hydro/VIC/releases/tag/Mao_etal_stateDA_May2018). The DA code used in this study is
available at https://github.com/UW-Hydro/dual_DA_SMAP.

**Author contribution**

All co-authors designed the experiments. Yixin Mao developed the system code and

carried out the experiments. Wade T. Crow and Bart Nijssen supervised the study. Yixin Mao
prepared the manuscript with contributions from all co-authors.

**Competing interests**

The authors declare that they have no conflict of interest.


## Acknowledgements

This work was supported in part by NASA Terrestrial Hydrology Program Award

NNX16AC50G to the University of Washington and NASA Terrestrial Hydrology Program
Award 13-THP13-0022 to the United States Department of Agriculture, Agricultural Research
Service. Yixin Mao also received a Pathfinder Fellowship by CUAHSI with support from the
National Science Foundation (NSF) Cooperative Agreement No. EAR-1338606. We would also
like to thank Andrew Wood from NCAR for help on calibration.

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

**Table 1.** List of USGS streamflow sites used for verification.

| Basin number | USGS station no. | USGS station name | Short name |
|---|---|---|---|
| 1 | 07144200 | Little Arkansas River at Valley Center, KS | L Arkansas |
| 2 | 07144780 | Ninnescah River AB Cheney Re, KS | Ninnescah |
| 3 | 07147800 | Walnut River at Winfield, KS | Walnut |
| 4 | 07152000 | Chikaskia River near Blackwell, OK | Chikaskia |
| 5 | 07177500 | Bird Creek Near Sperry, OK | Bird |
| 6 | 07186000 | Spring River near Wace, MO | Spring |
| 7 | 07196500 | Illinois River near Tahlequah, OK | Illinois |
| 8 | 07243500 | Deep Fork near Beggs, OK | Deep |



**Table 2.** Daily streamflow results from the dual correction system for the eight USGS sub-basins
shown in Fig. 1. In addition to the deterministic KGE improvement, PER and probabilistic
NENSK results from the dual system ("*dual*" columns), the table also lists the open-loop
streamflow KGE ("*open-loop KGE*" column), KGE improvement and PER as a result of state
update or rainfall correction scheme alone ("*state update only*" and "*rainfall correction only*"
columns, respectively), and KGE improvement and PER when forced by the NLDAS-2
benchmark precipitation without state update ("*NLDAS-2 forced*" column).

| | Open-loop KGE | KGE improvement | | | | PER | | | | NENSK |
|---|---|---|---|---|---|---|---|---|---|---|
| | | Dual | State update only | Rainfall correction only | NLDAS2-forced | Dual | State update only | Rainfall correction only | NLDAS2-forced | Dual |
| L Arkansas | -0.12 | +0.17 | +0.23 | -0.01 | +0.57 | 7.3% | 10.8% | 1.2% | 40.0% | 1.98 |
| Ninnescah | 0.25 | +0.15 | +0.06 | +0.16 | +0.20 | 14.0% | 5.5% | 13.7% | 30.4% | 0.35 |
| Walnut | 0.54 | -0.02 | -0.03 | +0.03 | -0.23 | 5.8% | 5.7% | 2.8% | 23.3% | 2.70 |
| Chikaskia | 0.67 | +0.07 | +0.05 | +0.02 | -0.45 | 15.0% | 11.1% | 6.6% | 2.2% | 1.96 |
| Bird | -1.49 | +0.95 | +0.58 | +0.63 | +0.95 | 33.5% | 17.0% | 25.8% | 68.9% | 2.01 |
| Spring | -3.64 | +0.83 | +0.65 | +0.33 | +3.93 | 13.2% | 8.7% | 7.0% | 83.4% | 13.11 |
| Illinois | -1.91 | +0.50 | +0.36 | +0.26 | +2.72 | 17.6% | 7.4% | 12.9% | 81.8% | 13.78 |
| Deep | -0.77 | +0.49 | +0.39 | +0.37 | +1.55 | 20.8% | 13.1% | 21.2% | 68.3% | 2.34 |



**Table 3.** Review of SMART rainfall correction results in literature along with the results in this
study.

| Literature | Baseline rainfall product | Benchmark rainfall product | SM product | Domain | Accumulation period | Baseline correlation $r$ | $r$ improvement | Baseline RMSE (mm) | PER |
|---|---|---|---|---|---|---|---|---|---|
| Crow et al. (2009) | TRMM 3B40RT | CPC rain gauge analysis | AMSR-E | Southern Great Plain | 3-day | ~ 0.5 | ~ + 0.2 | 13.0 | ~ 30% |
| | | | | CONUS | 3-day | ~ 0.55 | ~ + 0.05 | 11.8 | ~ 15% |
| Crow et al. (2011) | TRMM 3B40RT | CPC rain gauge analysis | AMSR-E | CONUS | 3-day | ~ 0.55 | ~ + 0.1 | 13.1 | ~ 20% |
| Chen et al. (2012) | Princeton Global Forcing Dataset | CPC rain gauge analysis | SMMR, SMM/I, ERS | Global | 10-day | ~ 0.35 | ~ + 0.15 | - | - |
| Brocca et al. (2016) | TRMM 3B42RT | AWAP rain gauge product | SMOS | Australia | 1-day | 0.62 | +0.01 | 5.6 | 7% |
| | | | | | 5-day | 0.71 | +0.05 | 14.0 | 14% |
| This study | IMERG Early Run | NLDAS-2 | SMAP L3 Passive | Arkansas-Red | 1-day | 0.80 | +0.02 | 6.1 | 8% |
| | | | | | 3-day | 0.82 | +0.02 | 11.0 | 8% |


