# Peer review of "Dual state/rainfall correction via soil moisture assimilation for improved streamflow simulation: Evaluation of a large-scale implementation with SMAP satellite data"

_Hydrology and Earth System Sciences, 2019_

## Referee Comment (RC1) · Christian Massari (Referee) · 19 Apr 2019

Review of the manuscript:

Dual state/rainfall correction via soil moisture assimilation for improved streamflow simulation: Evaluation of a large-scale implementation with SMAP satellite data.

by Mao et al.

The manuscript deals with the assimilation of satellite soil moisture observations de-

rived from the SMAP mission for improving streamflow simulations in the Arkansas-Red River basin. The ingestion of SMAP satellite soil moisture data into the system is carried via the so called "Dual state/rainfall correction" a method already used by one of the authors in other studies published in recent years. With respect to previous studies the authors used a satellite precipitation estimates derived from the new Global Precipitation Measurement mission (GPM), i.e., the IMERG Early run product which is the near real time version of the three available IMERG products. With respect to the products available in the past like those derived from the TRMM mission this satellite product has increased the accuracy and detection skill thanks to a new Dual Precipitation radar with light rainfall detection capabilities. Therefore one of the research question is whether satellite soil moisture observations are still able to improve the quality of the rainfall time series (and also the model states) in a way that it can be beneficial for streamflow simulations. In particular, satellite soil moisture observations are derived from the recent SMAP mission which has demonstrated to release products with a relatively high signal to noise ratio.

The manuscript is well written and clear. It is also of interest for the journal readership as the use of these new satellite products has been explored very little due to the relatively short observation period. The focus on this "dual correction" scheme is also very interesting as the improvement of streamflow simulations can be obtained either via the correction of states (in this case several an improvement of low flows is obtained with respect to high flows) or via correction of rainfall (which seems to have a benefit on high flows, from previous studies cited by the authors).

I have only one major comment which is related to the rainfall correction and its effect on the streamflow simulations which to me is a bit ambiguous and should be improved. In many parts of the manuscript it is said that the correction of the rainfall has a smaller effect since the rainfall forcing used (IMERG-ER) has a good quality (see lines 331 onward). However, this contradict with the results in Table 3 where the open loop simulations show in some cases of very poor performance of flood simulations ( which

are likely due the poor rainfall quality) and with other sentences stating that the IMERG-ER has large errors (line 448) in some basins. In fact, when forced by NLDAS2 there is a significant increase of the model performance up to 80% of PER which however, is still not satisfactory for some basins (see Table 3 Walnut, Chikaskia and Spring).

Then, I think there two possible reasons. Either SMAP adds little in terms of rainfall correction or SMART only corrects for the random error component which is the component the hydrological model is less sensitive to as correctly stated by the authors. Therefore the systematic error can be very important in this respect. However as between the two precipitation products it is difficult to judge which one is really better (at least by looking at the performance in terms of KGE) I suggest to compare them with a gauge-based dataset like CPC or Stage IV both in terms of rainfall (bias, correlation and error) and in terms of streamflow simulations. Indeed it is well known that these two products works really well in US (see for example the last study of Beck et al. 2019 where they used Stage IV as a reference for validating precipitation products over CONUS). Another solution could be to drive VIC model with IMERG final run which is corrected with rain gauge and therefore should have a lower bias with its near real time counterpart and thus would explain if the systematic error is the real problem. To summarize my suggestion is to include in the study a reference precipitation product against to compare IMERG-ER and NLDAS2. That would shed some light on the problems of the poor performance simulations.

Based on that, my suggestion is MODERATE revisions of the manuscript.

Christian Massari

Beck, H. E., Pan, M., Roy, T., Weedon, G. P., Pappenberger, F., van Dijk, A. I., ... & Wood, E. F. (2019). Daily evaluation of 26 precipitation datasets using Stage-IV gauge-radar data for the CONUS. Hydrology and Earth System Sciences, 23(1), 207-224.

---

## Referee Comment (RC2) · Anonymous Referee #2 · 21 Jun 2019

In this work, authors have attempted to carry out assimilation of SMAP soil moisture to correct rainfall using SMART algorithm and soil moisture state of VIC model using Ensemble Smoother over Arkansas-Red sub-basins.

The topic is of interest to hydrological community and the some of the conclusions made are important. However, the quality of writing not up the standard of HESS. As the authors have acknowledged that the methods used in this paper have already been implemented elsewhere in the literature, and the only "new" contribution is in terms of using new datasets, there should have been deeper discussion and analysis regarding

the outcome of this experiment. I agree that authors have used Ensemble Smoother as an extension to EnKF in this work. However, the results suggest improvements only when the updates are made at coarse temporal scale (SMAP scale). So, apart from minor differences, there may not be statistically significant difference in terms of performance between the two techniques. I will be glad if I am proven otherwise. Also, most importantly, there was only a speculative attribution of lack of improvement in performance to the better quality of IMERG precipitation. The results lack appropriate robust quantitative analysis in this regard. Further comments are listed below. In summary, the manuscript may have to be revised thoroughly in a wat that highlights the major contributions, and also show how these contributions are helping us to extend our understanding in this domain of research. In this process, please also consider addressing the following specific and minor comments:

Specific Comments:

1) SMAP soil moisture estimates have a maximum sensing depth upto 6 cm in vegetated areas (Babaeian et al., 2019, Reviews of Geophysics). The deeper soil moisture has stable temporal dynamics compared with that of surface soil moisture. Further, the VIC model executed at 10, 40 and 93 cm. In the process of assimilation, the SMAP soil moisture are rescaled to VIC soil moisture dynamics. So, essentially the noisier timeseries (surface soil moisture) is being rescaled using the temporal dynamics of smoother timeseries (VIC soil moisture). Can authors assess the implications of this mismatch on the final outcome?

2) How is the soil moisture state in the deeper layers being updated? Is there a correction factor implemented here, as carried out by Lievens et al. (2015, 2016)? Although authors have mentioned in Line 221, an equation will bring clarity to their statement.

3) Equations will help to understand the mathematically involved procedure like data assimilation.

4) Authors may have to discuss the sensitivity of choosing gamma parameter in Eq.

(1).

5) L 210: There is also a need for authors to explain why the error variance of 0.3 mm2 is chosen and its sensitivity.

6) L: 228: When only top two layers are being updated, why is it that all the three layers are perturbed?

7) L: 230-233: I find that this statement in qualitative in nature. So, it cannot be considered as a finding.

8) Figure 3 is not explained properly. What is the meaning of improvement in correlation? Is it correlation(NLDAS, IMERG_Corrected) – correlation(NLDAS, IMERG_Original)? There is no detail about it in the manuscript.

9) L: 302: If delta and P are aggregated to 3-day windows prior to correction in the case of EnKF, why are there minor changes in the spatial maps in Fig. 3 (d-f)? Will it not be sensible to just have a 3-day window map?

10) Interestingly, there seems to be an overlap in the spatial patterns of Figs. 2 and 3. It appears that there is a correlation improvement in the western part, which received lower rainfall compared to the eastern region. Is there such dependence of rainfall amount on the performance of correction?

11) I think it will be better if bias and error maps are also plotted to comprehensively characterize the errors.

12) L: 333-334: This is one of the most important statements made by authors. I think it is important to support this statement with rigorous analysis. I think it may not be fair to compare these results with that of Table 2. This is because of a) the experimental setup has changed, b) case study has changed, and c) the reference dataset has changed.

13) Figure 3: Since there is a median correlation improvement difference of only 0.01, can't we just use EnKF, which is much simpler compared to Ensemble Smoother?

14) Figure 4: It is understandable that in the case of correcting rainfall at all timesteps, SMART can misinterpret SM retrieval noise as small rainfall corrections. Can this issue be alleviated by considering a threshold of, say 2 mm to classify rain/no-rain and continuously correct the rainfall. This way the SM retrieval noise can still be pushed to zero, and there may some reduction of uncertainty due to rain/no-rain classification.

15) L: 318 is a speculative statement with no strong analysis.

16) Section 3.1.2: (in alignment with my Comment 12) I think correlation may not be sufficient to conclude on the quality of rainfall product. There can be other forms of error (such as bias), which are not being considered in this analysis.

17) Authors should provide some insights into the spatial patterns in Fig. 5. If median value is all that is needed in the discussion, then what is the need to have such spatial maps?

18) Section 3.2.1: I think there is a need to compare the rainfall products with a third product to get a complete picture of relative errors between the products.

19) There is no discussion regarding Figs. 6 and 7 in the manuscript.

20) Fig. 7 Deep Site: Between June and July although there are spikes in the ensemble, why isn't there a peak in dual corrected time series (which is ensemble-mean)? Also, since these are unregulated catchments, any peak can be attributed to rainfall event. So, if there are spikes in the ensemble during this period, does it mean a) there is an anomalous rainfall or b) the assimilation technique erroneously updated the rainfall during this period? I think these streamflow timeseries should also contain rainfall timeseries to look at where the update is being carried out.

21) The discussion section is speculative not very convincing. Authors may have to carryout robust analysis to substantiate their findings.

Minor Comments:

22) Figure 4: the x-axis is not explained properly.

23) Abstract opens with statement that soil moisture is necessary for accurate stream-flow simulations. However, the conclusions are slightly contradictory. Please consider revising the abstract appropriately.

---

## Author Comment (AC1) · 19 Sep 2019

We appreciate the comments from the reviewers. We respond to each reviewer comment below:

 **Major comments: 1) I have only one major comment which is related to the rainfall correction and its effect on the streamflow simulations which to me is a bit ambiguous and should be improved. In many parts of the manuscript it is said that the correction of the rainfall has a smaller effect since the rainfall**

**forcing used (IMERG-ER) has a good quality (see lines 331 onward). However, this contradict with the results in Table 3 where the open loop simulations show in some cases of very poor performance of flood simulations (which are likely due the poor rainfall quality) and with other sentences stating that the IMERG-ER has large errors (line 448) in some basins. In fact, when forced by NLDAS2 there is a significant increase of the model performance up to 80**

We have responded the reviewer's comment via the following points: 1) We agree with the reviewer that we overstated the "good quality" of IMERG, since it is clear from our streamflow results that IMERG rainfall quality is not good in some sub-basins. To address this, we have toned down the argument that IMERG has "good quality", and instead emphasized that one reason of the smaller rainfall correction results than found by previous studies is because of the relatively better quality IMERG compared to older rainfall products (this discussion is now moved to Section 4.1 in the manuscript). In addition, the revised manuscript now clearly acknowledges (in Section 3.2.2) that in some sub-basins (the Bird, Spring, Illinois and Deep sub-basins in our experiment), SM-based rainfall correction scheme can potentially play an important role in improving VIC streamflow estimates because of relatively large IMERG error (with respect to the NLDAS-2 baseline). However, such potential improvement was not realized because these basins are densely vegetated with (subsequently) low SMAP quality. We believe that these revisions make our discussion more consistent, and balance and address the contradiction noted by the reviewer. 2) Regarding the addition of gauge-based rainfall dataset – the NLDAS-2 product used in the study is indeed already based on the gauge-based CPC rainfall (as well as ground radar), which is the reason that we used it as the reference precipitation in our study. Even if NLDAS-2 rainfall is not perfect especially when translating into streamflow results (as can be seen from our streamflow analysis), its reliance on gauge observations ensures that it is relatively more reliable than the other satellite-based rainfall products considered in this study. Therefore, it provides an adequate benchmark to evaluate the lower-quality satellite-based products. We have added a more detailed description of the NLDAS-2 rainfall

product in Section 2.2.4 to highlight these points.

---

## Author Comment (AC2) · 19 Sep 2019

**Response to reviewers**

**Dual state/rainfall correction via soil moisture assimilation for improved streamflow simulation: Evaluation of a large-scale implementation with SMAP satellite data**

**Yixin Mao, Wade T. Crow, Bart Nijssen**

**Revision summary:**

We appreciate the comments from the reviewers. We respond to each reviewer comment below, with reviewer comments shown in **bold**. We have also made minor edits throughout the manuscript to make it more succinct and readable.

**Response to Reviewer 1 (Christian Massari)**

**Major comments:**

**1) I have only one major comment which is related to the rainfall correction and its effect on the streamflow simulations which to me is a bit ambiguous and should be improved. In many parts of the manuscript it is said that the correction of the rainfall has a smaller effect since the rainfall forcing used (IMERG-ER) has a good quality (see lines 331 onward). However, this contradict with the results in Table 3 where the open loop simulations show in some cases of very poor performance of flood simulations (which are likely due the poor rainfall quality) and with other sentences stating that the IMERG-ER has large errors (line 448) in some basins. In fact, when forced by NLDAS2 there is a significant increase of the model performance up to 80% of PER which however, is still not satisfactory for some basins (see Table 3 Walnut, Chikaskia and Spring). Then, I think there two possible reasons. Either SMAP adds little in terms of rainfall correction or SMART only corrects for the random error component which is the component the hydrological model is less sensitive to as correctly stated by the authors. Therefore the systematic error can be very important in this respect. However as between the two precipitation products it is difficult to judge which one is really better (at least by looking at the performance in terms of KGE). I suggest to compare them with a gauge-based dataset like CPC or Stage IV both in terms of rainfall (bias, correlation and error) and in terms of streamflow simulations. Indeed it is well known that these two products works really well in US (see for example the last study of Beck et al. 2019 where they used Stage IV as a reference for validating precipitation products over CONUS). Another solution could be to drive VIC model with IMERG final run which is corrected with rain gauge and therefore should have a lower bias with its near real time counterpart and thus would explain if the systematic error is the real problem. To summarize my suggestion is to**

**include in the study a reference precipitation product against to compare IMERG-ER and NLDAS2. That would shed some light on the problems of the poor performance simulations.**

We have responded the reviewer's comment via the following points:

1) We agree with the reviewer that we overstated the "good quality" of IMERG, since it is clear from our streamflow results that IMERG rainfall quality is not good in some sub-basins. To address this, we have toned down the argument that IMERG has "good quality", and instead emphasized that one reason of the smaller rainfall correction results than found by previous studies is because of the *relatively* better quality IMERG compared to older rainfall products (this discussion is now moved to Section 4.1 in the manuscript). In addition, the revised manuscript now clearly acknowledges (in Section 3.2.2) that in some sub-basins (the Bird, Spring, Illinois and Deep sub-basins in our experiment), SM-based rainfall correction scheme can potentially play an important role in improving VIC streamflow estimates because of relatively large IMERG error (with respect to the NLDAS-2 baseline). However, such potential improvement was not realized because these basins are densely vegetated with (subsequently) low SMAP quality. We believe that these revisions make our discussion more consistent, and balance and address the contradiction noted by the reviewer.

2) Regarding the addition of gauge-based rainfall dataset – the NLDAS-2 product used in the study is indeed already based on the gauge-based CPC rainfall (as well as ground radar), which is the reason that we used it as the reference precipitation in our study. Even if NLDAS-2 rainfall is not perfect especially when translating into streamflow results (as can be seen from our streamflow analysis), its reliance on gauge observations ensures that it is relatively more reliable than the other satellite-based rainfall products considered in this study. Therefore, it provides an adequate benchmark to evaluate the lower-quality satellite-based products. We have added a more detailed description of the NLDAS-2 rainfall product in Section 2.2.4 to highlight these points.

**Response to Reviewer 2**

**The topic is of interest to hydrological community and the some of the conclusions made are important. However, the quality of writing not up the standard of HESS. As the authors have acknowledged that the methods used in this paper have already been implemented elsewhere in the literature, and the only "new" contribution is in terms of using new datasets, there should have been deeper discussion and analysis regarding the outcome of this experiment. I agree that authors have used Ensemble Smoother as an extension to EnKF in this work. However, the results suggest improvements only when the updates are made at coarse temporal scale (SMAP scale). So, apart from minor differences, there may not be statistically significant difference in terms of performance between the two techniques. I will be glad if I am proven otherwise. Also, most importantly, there was**

only a speculative attribution of lack of improvement in performance to the better quality of IMERG precipitation. The results lack appropriate robust quantitative analysis in this regard. Further comments are listed below. In summary, the manuscript may have to be revised thoroughly in a wat that highlights the major contributions, and also show how these contributions are helping us to extend our understanding in this domain of research. In this process, please also consider addressing the following specific and minor comments:

**Major comments:**

**1) SMAP soil moisture estimates have a maximum sensing depth up to 6 cm in vegetated areas (Babaeian et al., 2019, Reviews of Geophysics). The deeper soil moisture has stable temporal dynamics compared with that of surface soil moisture. Further, the VIC model executed at 10, 40 and 93 cm. In the process of assimilation, the SMAP soil moisture are rescaled to VIC soil moisture dynamics. So, essentially the noisier timeseries (surface soil moisture) is being rescaled using the temporal dynamics of smoother timeseries (VIC soil moisture). Can authors assess the implications of this mismatch on the final outcome?**

We agree with the reviewer that matching a satellite-observed soil moisture product with that represented in a land surface model (LSM) is a very challenging task, and so far there is no standard good solution despite many research efforts (see, e.g., Yilmaz and Crow, 2013; Kumar et al., 2015; Nearing et al., 2018). On the one hand, although SMAP is typically described as measuring the top ~ 5 cm of soil moisture, the actual vertical support depth is unclear and varies nonlinearly as a function of soil moisture and vegetation water content. On the other hand, the relationship between the top-layer depth and its soil moisture dynamics in an LSM is complex and driven by a large number of poorly known model parameters (although, Shellito et al. (2018) found that changing the top-layer depth from 10 cm to 5 cm in the Noah LSM did not affect surface soil moisture dynamics much). Therefore, contrasts between the spectral characteristics of modelled and observed "surface" soil moisture is a general problem for essentially all land data assimilation systems (Qiu et al., 2014) – even those in which a concerted effort is made to "match" the vertical support of both estimates. While it likely does introduce some time scale error, the moment-matching rescaling techniques as used in our study is one of the standard, although imperfect, solutions, which is commonly used in soil moisture data assimilation studies. Therefore, we have kept our original procedure and added new discussion in Section 4.2 which acknowledges this shortcoming.

**2) How is the soil moisture state in the deeper layers being updated? Is there a correction factor implemented here, as carried out by Lievens et al. (2015, 2016)? Although authors have mentioned in Line 221, an equation will bring clarity to their statement.**

In our 3-layer VIC setup, the middle layer is updated using the surface measurement via a standard EnKF algorithm (i.e., perturbed and updated based on the error covariance calculated based on the ensemble distribution) – this follows the approach of Lievens et al. (2016) but differs slightly from Lievens et al. (2015) where an artificial vertical correlation factor was used to "nudge" the deeper-layer state. For the bottom layer, we did not include it in the EnKF update, which is the same as in Lievens et al. (2015, 2016) and further justified by Mao et al. (2019). We still perturbed the bottom layer to create a realistic ensemble estimate. All these modeling choices were detailed earlier in Mao et al. (2019), and now clarified in the revised text (and with additional equations in Supplemental material).

**3) Equations will help to understand the mathematically involved procedure like data assimilation.**

We have added the key equations and descriptions in Supplemental Material as suggested.

**4) Authors may have to discuss the sensitivity of choosing gamma parameter in Eq. (1).**

First, we would like to emphasize that the gamma parameter in Equation (1) was already manually tuned with the objective of maximizing the correlation coefficient between the uncorrected API time series and the SMAP time series over the domain, such that the API model as stated in Equation (1) captures the SMAP-observed SM dynamics as much as possible. In addition, this issue has been examined in past studies. Using a very similar system, Crow et al. (2011) found that the magnitude of rainfall correction was minimally sensitive to variations in gamma in the effort of mimicking a more complex soil water balance model.

Second, we have added a sensitivity analysis to examine the impact of gamma on rainfall correction results, as suggested by the reviewer. Figures 1 and 2 below show the domain-median correlation coefficient improvement and percent RMSE reduction (PER), respectively, after correction at different gamma values (in the manuscript, gamma = 0.98 was used). We see that around the chosen gamma = 0.98, the sensitivity of rainfall correction performance to gamma is relatively small, and gamma = 0.98 results in optimal PER when evaluating at 1-day and 3-day time steps (although performance is even better at gamma = 0.99 for the other measures shown). However, we also see that the correction performance is significantly degraded if gamma is far from the chosen value (i.e., if gamma < 0.95). These results should confirm that the chosen gamma value in the manuscript is reasonable and roughly optimal. This analysis is now presented in the revised Supplemental Material.

[Figure]

**Figure 1.** Domain-median correlation coefficient improvement of IMERG rainfall after SMART correction (with respect to the NLDAS-2 reference) at different γ values. The improvement is evaluated for 3-hour (3H), 1-day (1D) and 3-day (3D) accumulation intervals.

[Figure]

**Figure 2.** Same as Figure 1, but evaluated by percent RMSE reduction (PER).

**5) L 210: There is also a need for authors to explain why the error variance of 0.3 mm2 is chosen and its sensitivity.**

According to the Kalman filter theory, the time series of the normalized filter innovation should have mean zero and variance one. The normalized filter innovation, $e$, is defined as

$$e_k = \frac{\tilde{y}_k - \tilde{y}_k^-}{\sqrt{H_k P_k^- H_k^T + R_k}} \tag{1}$$

where $k$ is the time step index, $\tilde{y}$ is the measurement, $\tilde{y}^-$ is the estimated measurement before update, $H$ is the vector mapping from state to measurement space, $P^-$ is the estimated state error covariance, and $R$ is the measurement error variance. Since we have a relatively good estimate of measurement error, the only degree of freedom to tune the innovation variance is the state error level, for which 0.3 mm$^2$ was found to roughly satisfy the statistical requirement on the filter innovation. Since the innovation is required to have these statistical properties by the Kalman filter theory, this is not something that can be freely altered and we did not carry out a sensitivity analysis. This point has been clarified in the revised text.

**6) L: 228: When only top two layers are being updated, why is it that all the three layers are perturbed?**

While the perturbation of the bottom layer does not affect the EnKF updating procedure, we need to perturb the bottom layer to generate a realistic ensemble for it since we are interested in probabilistic streamflow estimation (and the bottom layer soil moisture impacts VIC streamflow estimates via its role in determining baseflow). While ensemble spread in the first two soil layers will eventually propagate into the (third) bottom layer, such spread does not explicitly account for errors that originate in the bottom layer. We have clarified it in the text.

**7) L: 230-233: I find that this statement in qualitative in nature. So, it cannot be considered as a finding.**

We did carry out the experiment of comparing the state update performance with and without considering the spatial auto-correlation of states, and found that considering spatial autocorrelation did not improve EnKF result (detailed results not shown). We have clarified this in the revised text.

**8) Figure 3 is not explained properly. What is the meaning of improvement in correlation? Is it correlation (NLDAS, IMERG_Corrected) – correlation(NLDAS, IMERG_Original)? There is no detail about it in the manuscript.**

Figure 3 shows the improvement of the IMERG rainfall product relative to the NLDAS-2 reference before and after the SMART rainfall correction - the formula written out by the reviewer is correct. However, we have decided to leave out this formula to avoid extra notation, but instead added a clearer description in the caption.

**9) L: 302: If delta and P are aggregated to 3-day windows prior to correction in the case of EnKF, why are there minor changes in the spatial maps in Fig. 3 (d-f)? Will it not be sensible to just have a 3-day window map?**

Even if EnKF corrects the 3-day accumulated rainfall amounts, the 3-day rainfall delta is downscaled uniformly to every 3-hour time step under the 3-day window. Therefore, the 3-hourly (or daily) rainfall can still be improved to be closer to truth, even if the correction does not capture the fine temporal resolution. We have clarified this in the revised text.

**10) Interestingly, there seems to be an overlap in the spatial patterns of Figs. 2 and 3. It appears that there is a correlation improvement in the western part, which received lower rainfall compared to the eastern region. Is there such dependence of rainfall amount on the performance of correction?**

We have added discussion on the spatial pattern of rainfall correction as suggested by the reviewer (first paragraph of Section 3.1.2). Specifically, SMAP tends to have better quality (in terms of correlation improvement) in the western part of the domain due to less vegetation, which is one possible reason that it adds more value to the SMART rainfall correction in the western region. RMSE is reduced more in the eastern part of the domain, which is likely due to the better correction for larger rainfall events (which mostly happen in the east).

**11) I think it will be better if bias and error maps are also plotted to comprehensively characterize the errors.**

The error (in terms of RMSE) reduction map was already included in the manuscript (Figure 5, left column). We do not include a bias correction map since the SMART algorithm does not correct overall rainfall bias – it rescales the corrected time series back to have the same mean as the uncorrected time series (this is pointed out in the first paragraph of Section 3.1.1).

**12) L: 333-334: This is one of the most important statements made by authors. I think it is important to support this statement with rigorous analysis. I think it may not be fair to compare these results with that of Table 2. This is because of a) the experimental setup has changed, b) case study has changed, and c) the reference dataset has changed.**

First, we have toned down the argument that IMERG has "good quality", and instead emphasized the main reason for the smaller rainfall correction results than those found by previous studies is the *relatively* better quality IMERG compared to older rainfall products. We have also pointed out that SMAP's quality is low in dense-biomass regions, which limits its ability to correct IMERG rainfall. Therefore, the revised manuscript now relies less heavily on this argument to explain key results.

Nevertheless, the tendency for marginal data assimilation improvement to decrease as the skill of the background increases is a very well-developed *general* concept in land data assimilation (Reichle et al., 2008; Qing et al., 2011; Bolten and Crow, 2012; Dong et al., 2019) – and has already been demonstrated for the specific case of using soil moisture to correct rainfall (Crow et al., 2011). In addition, Crow and Ryu (2009) already provided exactly the rigorous analysis requested by the author. That is, using a conceptually equivalent rainfall correction approach and a set of well-controlled synthetic experiments, they examined the impact of baseline precipitation analysis on marginal precipitation skill improvements. Their conclusions (also) clearly demonstrate that rainfall correction margins are degraded by improvements in baseline precipitation skill (i.e., the exact point made here). Finally, while the approaches applied in Table 2 differ slightly, it should be noted that various correction approaches (e.g. the SM2RAIN used by Brocca et al. (2013, 2014) and the SMART approach applied by Crow et al. (2011)) have been inter-compared and found to perform similarly (Brocca et al., 2016), suggesting that their results are fairly cross-comparable (as in Table 2). Therefore, our hypothesis here is supported by a range of earlier studies and a well-demonstrated concept in land data assimilation. We have clarified these points in Section 4.1 in the revised manuscript.

**13) Figure 3: Since there is a median correlation improvement difference of only 0.01, can't we just use EnKF, which is much simpler compared to Ensemble Smoother?**

First of all, the EnKS is not really more complicated or computationally demanding than the EnKF. As a result, there is no significant downside to use the EnKS instead of an EnKF. Secondly, since the baseline correlation coefficent of IMERG is already quite good (domain-median correlation coefficient above 0.8 relative to NLDAS-2 reference), it is a relatively difficult task to further improve it, and even small correlation improvements are significant (in the context of remaining unexplained variability). Finally, the correlation improvement achieved by EnKS is also much more obvious in certain parts of the domain (e.g., western end; see Figure 3) compared to that by EnKF, despite the relatively small difference in domain-median improvement.

**14) Figure 4: It is understandable that in the case of correcting rainfall at all timesteps, SMART can misinterpret SM retrieval noise as small rainfall corrections. Can this issue be alleviated by considering a threshold of, say 2 mm to classify rain/no-rain and continuously correct the rainfall. This way the SM retrieval noise can still be pushed to zero, and there may some reduction of uncertainty due to rain/no-rain classification.**

We have added a sensitivity analysis as suggested by the reviewer. Specifically, we alter the threshold of classifying IMERGE rain/no rain (this threshold is essentially set to zero in the original manuscript, and SMART only corrects time steps during which rainfall occurs), and observe its impact on the rainfall correction results (i.e., categorical metrics at different rainfall scales as well as correlation improvement and percent RMSE reduction (PER)).

The following figures show the SMART correction results with different rain/no rain thresholds. For categorical metrics (Figure 1), having a rain/no rain threshold of 1 mm/3 hours or 2 mm/3 hours mitigates the issue of worsened POD at small rainfall events comparing to zero threshold, but also removes the (although small) FAR improvement. For mid-ranged rainfall events, a positive threshold mitigates the issue of worsened FAR as in the zero threshold case, but POD improvement becomes smaller. For larger rainfall events, POD improvement and TS improvement become slightly smaller (i.e., closer to zero) when using a positive rain/no rain threshold (note that the small positive rain/no rain threshold value can be considered as a "larger" rainfall event at some pixels with overall low precipitation, therefore affecting the categorical metrics toward the right side on the categorical metrics plots).

In addition to the categorical metrics, setting the rain/no rain threshold to either 1 mm/3 or 2 mm/3 hours slightly lowers values of correlation coefficient improvement and PER versus the baseline case of applying a rain/no rain threshold of zero accumulation (Figures 2 and 3).

In summary, there is no obvious optimized number for the rain/no rain threshold since there is a trade-off between POD and FAR. Although the overall TS at smaller rainfall events improves with a positive threshold, the correction for larger events, which are what SMART correction is more useful for, slightly worsens. A positive rain/no rain threshold does not benefit correlation coefficient and PER (which are sensitive to both POD and FAR performance). Based on these analyses, we have decided to keep the original analysis in the manuscript to have a zero rain/no rain threshold for SMART correction. We have added this sensitivity analysis to the revised Supplemental Material (and briefly mentioned key results of the analysis in the revised main text).

[Figure]

**Figure 3:** Change in categorical metrics (FAR, POD and TS) before and after SMART correction for 3-hourly, 1-day and 3-day accumulation periods. The left column (panels a, b and c) is the same as in Fig. 4 (right column) in the main text with SMART only correcting IMERG rainfall events with non-zero accumulation. The middle and right columns show the same metrics with SMART only correcting IMERG rainfall for events where accumulation rates exceed thresholds of 1 mm/3 hours and 2 mm/3 hours, respectively.

[Figure]

**Figure 4:** Correlation coefficient (with respect to the NLDAS-2 reference precipitation) improvement before and after SMART correlation for 3-hourly, 1-day and 3-day accumulation periods. As in Fig. 7, the left column (panels a, b and c) is the same as in Fig. 4 (right column) in the main text with SMART only correcting IMERG rainfall events with non-zero accumulation. The middle and right columns show the same metrics with SMART only correcting IMERG rainfall for events where accumulation rates exceed thresholds of 1 mm/3 hours and 2 mm/3 hours, respectively.

[Figure]

**Figure 5:** Same as Figure 4 above, but for percent RMSE reduction (PER; with respect to the NLDAS-2 reference precipitation). The left column (panels a, b and c) is the same as in Fig. 5 (left column) in the main text

**15) L: 318 is a speculative statement with no strong analysis.**

We have reworded the statement to list the improved rain/no rain detection of IMERG as one possible reason for the success of our tactic.

**16) Section 3.1.2: (in alignment with my Comment 12) I think correlation may not be sufficient to conclude on the quality of rainfall product. There can be other forms of error (such as bias), which are not being considered in this analysis.**

As mentioned above in Response to Reviewer 2 Major Comment 11, the original manuscript did include both an RMSE analysis (Figure 5, left column) as well as results based on a range of categorical metrics (e.g., POD, FAR and TS – see Figure 4) in the manuscript, with discussion in Section 3.1.2. We have added discussion of their spatial pattern in the revised test.

Overall bias is not designed to be corrected by the SMART algorithm and can therefore not be used as a metric for improvement (we have clarified this in Section 3.1.2).

**17) Authors should provide some insights into the spatial patterns in Fig. 5. If median value is all that is needed in the discussion, then what is the need to have such spatial maps?**

We have added discussion of the spatial pattern of the rainfall correction as suggested by the reviewer in Section 3.1.2. Specifically, RMSE is reduced more in the eastern part of the domain, which is likely due to the better correction for larger rainfall events (which mostly happens in the east). NENSK maps show that ensemble rainfall tends to be under-dispersed on the west edge of the domain with low rainfall, indicating that we are underestimating rainfall uncertainty in this region.

**18) Section 3.2.1: I think there is a need to compare the rainfall products with a third product to get a complete picture of relative errors between the products.**

As mentioned above in Response to Reviewer 1 Major Comments, NLDAS-2 precipitation is derived from daily gauge-based rainfall measurements and hourly ground-radar data, and is widely used. As a result, it is expected to be as generally reliable as any other ground-based rainfall product available in the region. Even if NLDAS-2 rainfall is not perfect (as can be seen from our streamflow results), its reliance on gauge observations ensures that it is relatively more reliable than the IMERG (and SMART-corrected) rainfall products considered in this study. Therefore, it provides an adequate benchmark for relative variation in skill and accuracy for these lower-quality products (we have added clarification on these in Section 2.2.4). We do not see any advantages of including an additional product for validation, particularly since that product will (inevitably) not be independent from NLDAS-2 (due to a shared dependence on common rain gauge datasets).

**19) There is no discussion regarding Figs. 6 and 7 in the manuscript.**

Figures 6 and 7 were discussed in Section 3.2.3 (the impact of VIC parameterization)..

**20) Fig. 7 Deep Site: Between June and July although there are spikes in the ensemble, why isn't there a peak in dual corrected time series (which is ensemble-mean)? Also, since these are unregulated catchments, any peak can be attributed to rainfall event. So, if there are spikes in the ensemble during this period, does it mean a) there is an anomalous rainfall or b) the assimilation technique erroneously updated the rainfall during this period? I think these streamflow timeseries should also contain rainfall timeseries to look at where the update is being carried out.**

As suggested by the reviewer, we have added rainfall data to the streamflow time series plot (the uncorrected IMERG rainfall (i.e., Figure 6) as well as the SMART-corrected rainfall ensemble (i.e., Figure 7). With the help of these (newly plotted) rainfall time series, the ensemble spikes at the Deep site between June and July (as an example) can be explained as follows:

1) For the spike around early July 2017: IMERG detected a small rainfall event, which correctly corresponded to a small rise in the gauge-observed streamflow. The ensemble of SMART-corrected rainfall is spread around the original IMERG time series without extreme peaks, but there are a few dual-corrected streamflow ensemble members with much-higher-than-observed spikes. This is likely because, given the hydrologic conditions during that time, 1) streamflow has a highly non-linear response to rainfall input in the VIC model, and/or 2) streamflow has a highly non-linear response to the SM state update in the VIC model.

2) For the spike around mid-June: the gauge-observed streamflow showed almost no spike at all while the uncorrected IMERG showed a small rainfall event, which indicates that this may be a false alarm event detected by IMERG. In this case, the few high-flow outlier ensemble members in the dual-corrected streamflow are likely due to both an inaccurate IMERG detection that is not successfully corrected by SMART, and the highly nonlinear response of streamflow to rainfall/SM state.

3) Because of the sometimes highly non-linear response of simulated streamflow to rainfall/state update, we plotted ensemble-median instead of ensemble-mean of the streamflow time series since the ensemble-median is a more stable representation of the "average" behavior of the streamflow ensemble. The ensemble-mean would, as the reviewer pointed out, bias toward a few outliers.

**Chikaskia**

[Figure]

all-time PER(RMSE) = 15.0%
all-time KGE_improve = 0.07
(open-loop KGE = 0.67)
all-time NENSK = 1.96

**Deep**

[Figure]

all-time PER(RMSE) = 20.8%
all-time KGE_improve = 0.49
(open-loop KGE = -0.77)
all-time NENSK = 2.34

**Illinois**

[Figure]

all-time PER(RMSE) = 17.6%
all-time KGE_improve = 0.50
(open-loop KGE = -1.91)
all-time NENSK = 13.78

[Figure]

Streamflow panel: ——— USGS observed ‒ ‒ ‒ Open-loop ——— Dual-corrected (light color for ensemble)

Precipitation panel: ‒ ‒ ‒ Uncorrected IMERG ——— Ensemble corrected rainfall

**Updated Figure 6.** Example time series of streamflow results from the dual correction system. In the lower panel, b*lack line*: USGS observed streamflow; *magenta line*: baseline VIC simulation; *light blue lines*: ensemble updated streamflow results; *solid blue line*: ensemble-mean updated streamflow. In the upper panel, *orange line*: uncorrected IMERG rainfall aggregated to the sub-basin-average; *light grey lines*: ensemble corrected rainfall. Only part of the simulation period is shown for clear display; however, statistics shown on each panel are based on the entire simulation period (approximately 2.5 years).

[Figure]

**Chikaskia**

(a)

all-time PER(RMSE) = 10.7%
all-time KGE_improve = 0.01
(open-loop KGE = 0.78)
all-time NENSK = 2.60

[Figure]

**Deep**

(b)

all-time PER(RMSE) = 22.0%
all-time KGE_improve = 0.25
(open-loop KGE = 0.08)
all-time NENSK = 3.64

[Figure]

**Illinois**

(c)

all-time PER(RMSE) = 14.9%
all-time KGE_improve = 0.35
(open-loop KGE = -0.97)
all-time NENSK = 34.48

[Figure]

Streamflow panel: — USGS observed   - - - Open-loop   —— Dual-corrected (light color for ensemble)

Precipitation panel: - - - Uncorrected IMERG   —— Ensemble corrected rainfall

**Updated Figure 7.** Same as Figure 6, but calibrated VIC model parameters.

**21) The discussion section is speculative not very convincing. Authors may have to carry out robust analysis to substantiate their findings.**

We have re-organized our results and discussion sections to incorporate all the major comments from reviewers and streamlined our major findings. Specifically:

1) We have toned down the argument that IMERG has "good quality", and instead emphasized that the main reason for the smaller rainfall correction results than those found by previous studies is the *relatively* better quality IMERG compared to older rainfall products.

2) In addition, we pointed out that SMAP's quality is low in dense-biomass regions, resulting in underperformed SMART rainfall correction in such regions.

3) We have emphasized our finding that systematic error accounts for a significant fraction of the total streamflow error, and the systematic error cannot be corrected by Kalman-filter-based data assimilation techniques which aimed solely at reducing zero-mean random errors.

**Minor Comments:**

**22) Figure 4: the x-axis is not explained properly.**

We have added more description of the x-axis in the figure caption.

**23) Abstract opens with statement that soil moisture is necessary for accurate streamflow simulations. However, the conclusions are slightly contradictory. Please consider revising the abstract appropriately.**

We have reworded the first few sentences in the abstract. We also would like to point out that soil moisture probably still contains information to help simulate streamflow more accurately, but the findings of this study show that our current satellite measurement and data assimilation techniques are not fully extracting this information.

**References:**

[revised manuscript text omitted]